# Targeting Drug Chemo-Resistance in Cancer Using Natural Products

**DOI:** 10.3390/biomedicines9101353

**Published:** 2021-09-29

**Authors:** Wamidh H. Talib, Ahmad Riyad Alsayed, Muna Barakat, May Ibrahim Abu-Taha, Asma Ismail Mahmod

**Affiliations:** Department of Clinical Pharmacy and Therapeutic, Applied Science Private University, Amman 11931-166, Jordan; a_alsayed@asu.edu.jo (A.R.A.); m_barakat@asu.edu.jo (M.B.); m_abutaha@asu.edu.jo (M.I.A.-T.); asmamahmod1212@gmail.com (A.I.M.)

**Keywords:** anticancer natural products, drug efflux, drug detoxification, plants derived natural products

## Abstract

Cancer is one of the leading causes of death globally. The development of drug resistance is the main contributor to cancer-related mortality. Cancer cells exploit multiple mechanisms to reduce the therapeutic effects of anticancer drugs, thereby causing chemotherapy failure. Natural products are accessible, inexpensive, and less toxic sources of chemotherapeutic agents. Additionally, they have multiple mechanisms of action to inhibit various targets involved in the development of drug resistance. In this review, we have summarized the basic research and clinical applications of natural products as possible inhibitors for drug resistance in cancer. The molecular targets and the mechanisms of action of each natural product are also explained. Diverse drug resistance biomarkers were sensitive to natural products. P-glycoprotein and breast cancer resistance protein can be targeted by a large number of natural products. On the other hand, protein kinase C and topoisomerases were less sensitive to most of the studied natural products. The studies discussed in this review will provide a solid ground for scientists to explore the possible use of natural products in combination anticancer therapies to overcome drug resistance by targeting multiple drug resistance mechanisms.

## 1. Introduction

Cancer is the second most common cause of death after cardiovascular diseases. Statistics from the USA showed that the number of people diagnosed with cancer was 1.7 million in 2017 with 0.6 million deaths [1]. Ninety percent of deaths from cancer result from the development of drug resistance, which leads to the ineffectiveness of chemotherapeutic agents [2]. Drug resistance can be defined as the ability of cancer cells to reduce the potency and efficacy of chemotherapeutic agents [3]. In some types of cancer such as renal cancer and hepatocellular carcinoma, malignant cells start resistance without previous exposure to chemotherapeutic agents (intrinsic resistance) resulting in a poor response to initial treatment [4]. In other cases, cancer cells exhibit initial sensitivity to chemotherapy followed by a poor response due to the development of resistance (aquired resistance) [3]. Previous studies on cell lines and animal models revealed that drug resistance in cancer can be achieved by complex mechanisms, including drug efflux using ATP-binding cassette (ABC) transporter [5], altering the expression of proteins targeted by anticancer drugs [6], drug detoxification [7], augmenting repair mechanisms in DNA [8], and evasion of apoptosis [1]. The use of natural products in the treatment of diseases is very old. Historical documents show that the first use of natural products in medical treatment was reported in Mesopotamia and dates back to 2600 BC [9]. Extensive research was conducted to explore the potential of natural products in cancer therapy. These efforts resulted in the development of some effective drugs derived from natural phytochemicals [10]. Diverse approaches were tested to overcome drug resistance in cancer. However, natural products from medicinal plants and other natural sources represent a promising and cost-effective approach [11]. In this review, we summarize natural products that have the potential to overcome drug resistance in cancer. The target of each natural product was identified, and the mechanisms of action were discussed in experimental and clinical studies.

## 2. Drug Chemo-Resistance in Cancer: Mechanistic Bases

Nowadays, one of the most prominent challenges for cancer treatment is drug resistance as malignant cells persuade different mechanisms (Figure 1) to deviate from treatment and maintain their survival. Understanding these mechanisms may facilitate the development of novel drugs with new targeting strategies, which may have a promising clinical implication. In this part of the review, we will discuss several drug-resistance mechanisms that have clinical significance.

### 2.1. Drug Efflux 

One of the primary mechanisms for chemotherapy resistance is drug efflux, which is defined as drug transportation from the intracellular milieu using energy-dependent pumps [12,13]. High rates of drug efflux in the cancer cells reduce internal drug accumulation and potentiate the cell capability to escape from the treatment [14,15,16]. This phenomenon could be either intrinsic or acquired; in other words, it either already exists in the cell before or develops post-drug administration [1]. 

Sophisticated transmembrane transporters direct drug efflux, mainly the ATP-binding cassette (ABC) family [17]. In humans, there are 48 ABC transporters which are stratified into seven subdivisions (ABCA-ABCG) based on phylogenetic analysis [1,17,18]. They are involved in the exportation of endogenous substances, e.g., metabolites, vitamins and lipids, in addition to exogenous products such as toxins and drugs [18] Part of these transporters play a key role in acquiring multidrug resistance (MDR) characteristics to cancer chemotherapies, such as ABCB1, ABCC1, and ABCG2 [17].

ABCB1, also known as MDR1 or P-glycoprotein (P-gp), is one of the well-characterized transporters associated with drug resistance for several types of tumors such as leukemia and colorectal, kidney, and lung multiple myeloma [3,19,20]. For drug efflux, the cell is coupled with ATP hydrolysis and conformational changes in the transporter [21]. A wide variety of drugs can bind/be pumped through this transporter, such as vincristine, vinblastine, daunorubicin, doxorubicin, epirubicin, etoposide, paclitaxel, mitomycin c, and topotecan [22,23,24]. Thus, overexpression of ABCB1 potentiates cell competence to hamper the chemotherapy treatment [1].

ABCC1 overexpression, also known as a multidrug resistance-associated protein-1 (MRP1), plays a crucial role in the failure of chemotherapy in a number of malignant tumors, including prostate, breast, and lung cancers [22,23,24,25]. ABCC1 transporter can efflux different anticancer drugs such as anthracyclines, camptothecins, vinca alkaloids, a few kinase inhibitors, etoposide irinotecan, and methotrexate [26]. In addition, this type of transporter pumps organic anionic compounds, which are conjugated to either glutathione (GSH), glucuronide, or sulfate [27,28]. 

ABCG2 is recognized as a primary breast cancer-efflux transporter known as breast cancer resistance protein (BCRP) [29,30]. ABCG2 is expressed in CD133-positive cancer stem cells (CSCs) from human colorectal tumors; accordingly, it is considered a marker for such types of CSCs cancers [31]. Additionally, overexpression of this transporter was notified in other kinds of cancers such as acute myeloid leukemia, endometrial carcinoma, lung cancer, and melanoma [29,32]. ABCG2 is capable of translocating a variety of anticancer drugs, including positively and negatively charged drugs, including topoisomerase inhibitors, tyrosine kinase inhibitors, antimetabolites, flavopiridol (cyclin-dependent kinase inhibitor), JNJ-7706621 (CDK and aurora kinases inhibitor), and bicalutamide (non-steroidal anti-androgen) [29,30].

Furthermore, overexpression of ABCC2 and ABCC3 has a pivotal effect on the resistance of multiple cytotoxic drugs such as methotrexate, cisplatin, doxorubicin, and etoposide [33,34]. These were found to potentiate the drug resistance in breast, liver, and lung cancers [34,35,36]. Accordingly, a deep understanding of ABC transporters (structure, physiology, overexpression, and mutations) has a promising role in innovating clinically effective anticancer drugs.

### 2.2. Drug Detoxification

Drug detoxification is considered one of the prominent mechanisms to confront chemotherapy treatment. This process involves two main pathways. The first pathway (Phase I) is mediated by cytochrome P450 enzymes (CYP450), which encompasse hydrolysis and oxidation-reduction reactions [37,38]. The second pathway (Phase II) comprises primarily conjugation reactions, e.g., glutathionylation, glucuronidation, acetylation, methylation, and sulfonation reactions [39]. This phase is complimentary for the first pathway, as these reactions aim to enhance the hydrophilicity of the parent drug or Phase 1 metabolites in order to be excreted [15]. Moreover, ABC efflux transporters translocate the Phase II conjugated reactions outside the cell [27,28]. For instance, irinotecan (prodrug, topoisomerase-1 inhibitor) is metabolized in the liver by carboxylesterases to the active 7-ethyl-10-hydroxycamptothecin (SN-38). Then, SN-38 will be exposed to glucuronidation conjugation and active effluxing through ABC transporters [40]. Therefore, the synergistic activity between the detoxification mechanisms and efflux transporters significantly suppresses the chemotherapeutic effectiveness [41].

One major drug-resistance conjugation pathway is glutathionylation, which is mediated by the GSH-GST system [39]. GST, a family of enzymes that conjugate GSH to the chemo drugs, increased its hydrophilicity and could easily be effluxed out of the cell [27,28]. It has been reported that levels of GSH and GST increase proportionally with the cancer stage; however, there is interindividual variability between patients, which limits its clinical implication [42]. On the other hand, a positive correlation was observed between the level of the type GSTπ protein and cancer drug-resistance in variable neoplastic diseases [43,44,45,46]. At the gene level, there is a relationship between the GST gene polymorphism and tumor incidence [47] and the efficiency of chemotherapy [48,49].

Unfortunately, several chemotherapeutic drugs are substrates for detoxification processes. Therefore, focusing on the machinery of this area may help in overcoming the resistance problem.

### 2.3. Apoptosis Inhibition

Inhibition of cell death is a fundamental hallmark of cancer. Anticancer medications mainly target this mechanism by inducing programmed cell death, also called apoptosis [50]. Consequently, any alternation in the apoptotic machinery may contribute to drug resistance [15]. Apoptosis occurs through two main pathways: extrinsic and intrinsic [50]. Activation of the extrinsic pathway is mediated by the binding of the tumor necrosis factor family to their receptors on the cell surface, followed by activation of caspase-8, which will promote cell death initiation [51]. Stimulation of the intrinsic pathways is mitochondrially controlled by the imbalance between the pro (e.g., BAX and BAK) and anti-apoptotic proteins (e.g., BCL-2, BCL-XL, BCLw) [52,53]. Recruitment of proapoptotic signaling molecules primes the permeabilization of the mitochondrial outer membrane then triggers the release of cytochrome c and leads to activation of series apoptotic reactions via caspases [50].

The disparity between the pro and anti-apoptotic molecules also plays a role in inducing resistance against anticancer therapy [54]. Hence, cancer cells overexpress the anti-apoptotic proteins (such as Bcl-2, Akt, and NF-κB) and/or suppress or disturb the production of proapoptotic proteins such as BAX [54,55,56]. The positive association between the expression level of anti-apoptotic proteins and the ability of cancer cells to evade the treatment has been documented in different types of cancer such as breast cancer, acute myeloid leukemia, and non-Hodgkins lymphoma [57,58,59]. Presumably, elevated levels of Bcl-2 and Akt hinder cytochrome c release from the mitochondria; thus, the subsequent apoptotic cascade will be discouraged [56]. Activation of Akt is followed by NFκB phosphorylation, which impedes the apoptosis processes and promotes cancer survival. Both Akt and NFκB trigger Bcl-2 inhibitory activity and potentiate the cell resistance power [60]. In the clinical setting, the development of targeted therapy to control the pro or anti-apoptotic proteins may have a promising solution for cancer drug resistance and improve the clinical outcomes.

### 2.4. Enhanced DNA Damage Repair

Numerous chemotherapeutic drugs target DNA damage of the cancerous cells as the main mechanism of action, such as platinum-based drugs, alkylating agents, and anthracyclines [61]. However, this activity may be defended by cancer cell-DNA repair response, which reduces the drug efficacy and potentiates the resistance [62]. Multiple DNA repair mechanisms have been documented in the literature [62,63], including direct reversal, mismatch repair (MMR), nucleotide excision repair (NER), base excision repair (BER), homologous recombination (HR), and nonhomologous end joining (NHEJ). Many factors affect the pathway of the DNA restoration such as tissue location, nature of the DNA-drug adduct, and the involved proteins [15,64,65]. For instance, DNA repair endonuclease XPF and DNA excision repair protein ERCC1 play a vital role in the NER and inter-strand crosslink repair pathways [66]. Studies have shown a positive correlation between the overexpression of these proteins and the establishment of significant drug resistance, e.g., resistance for platinum-based drugs [67,68].

On the other hand, it has been reported that the mortality rate was significantly reduced in patients receiving cisplatin-based chemotherapy with negative-ERCC1 non-small cell lung and breast tumors compared to ERCC1- positive tumors [64,69]. Another example, resistance to the alkylating chemotherapeutic agents, was significantly linked with the overexpression of the O_6_-methylguanine DNA methyltransferase (MGMT) repair enzyme, as glioblastoma patients with increased levels of MGMT showed poor treatment outcomes and higher mortality rates compared with the patients with reduced expression levels [70]. Therefore, such proteins could be a prognostic marker and auspicious therapeutic target for many anticancer drugs.

### 2.5. Epigenetic Alterations

Besides the previous resistance pathways, one of the prominent mechanisms is epigenetic alterations. These alternations mainly affect the function and expression of the cell gene, rather than causing mutations in the DNA sequence [65,71]. Epigenetic alterations could be present in different ways, including DNA methylation patterns, histone modification, chromatin remodeling, and noncoding RNA related alterations [1,8].

DNA methylation is utilized during cell division by adding methyl-group covalently to DNA cytosine through DNA methyltransferases [72]. It has been reported that number of cancer genes are exposed to hypermethylation, which yields transcriptional silencing for the tumor suppressor genes (e.g., CpG promoter islands of tumor suppressor genes) [73,74]. For example, hypermethylation of gene promoters plays a pivotal role in the resistance of ovarian cancer cells towards cisplatin [73]. Conversely, demethylation or hypomethylation was known to affect the chemo-response of cancer cells and upregulate the expression of oncogenes. For instance, hypomethylation of the ABCB1 promoter leads to overexpression of the efflux ABCB1 transporter, which potentiates drug resistance in esophageal squamous cancer cells [75]. Another study has revealed that DNA demethylation and histone modification at the promoter region enhances the overexpression of protein thymosin β4 (Tβ4), which contributes to drug resistance in hepatocellular carcinoma (HCC) cell line to VEGFR inhibitor sorafenib [76]. A study conducted by Bhatla et al. has demonstrated that suppression of DNA methylation and histone modification in acute lymphoblastic leukemia cells reverse the disease relapse and restore the cell chemosensitivity [77]. Therefore, defeating these resistance mechanisms may have a promising contribution in cancer therapy, as found in the management of resistant–heterogenous multiple myeloma [78].

Moreover, epigenetic modifications could also be present as chromatin remodeling and noncoding RNA-related alterations, including microRNAs (miRNAs) and long noncoding RNAs (lncRNAs) [79,80]. MiRNAs modulate the post-transcriptional gene expression and protein synthesis [81]. LncRNAs regulate gene expression through chromatin modification and hinder transcription activation [79,80]. Both noncoding RNAs affect the contribution to chemoresistance through modulation of protein production. Various studies have demonstrated overexpression and oncogenic activity of miRNA and lncRNA in different types of cancer such as lymphoma, lung, breast, stomach, colon, and pancreatic cancer [81,82,83]. Then again, these epigenetic alternations could be considered a futuristic target and have a role in cancer hallmarks.

### 2.6. ATP-Mediated Drug Resistance

Resistance to chemotherapy is also induced by ATP-mediated pathways, either intracellularly or extracellularly. Studies have shown that the intracellular level of ATP in malignant cells is usually more than healthy cells of the same source. In fact, that elevation in intracellular-ATP is mainly caused by increased glycolytic metabolism in a pathway called the Warburg effect [84]. This effect is considered a hallmark in approximately all cancer types [85,86]. Additionally, it was reported that drug-resistant cancer cells exhibit higher levels of intracellular ATP than the other tumor cells from the same tissue, which are required for cell survival under cytotoxic conditions [87,88]. For example, a study conducted by Zhou et al. has demonstrated that chemo-resistant colon cancer cell lines express higher levels (i.e., double) of intracellular ATP than non-resistant cells [87]. Contrarywise, sensitivity to chemotherapy was enhanced by diminishing intracellular ATP levels and suppressing the glycolysis process in the resistant cells (i.e., glycolysis, 3-bromopyruvate) [87].

Moreover, cancer cells are capable of extensively uptaking the extracellular ATP, subsequently increasing the intracellular ATP, and potentiating the cells’ tendency for drug resistance and cancer cell survival [89]. In many cancer types, the extracellular ATP was 10^3^ to 10^4^ times more than the normal cells from the exact origin [89,90]. Studies have shown that the uptake of extracellular ATP can be utilized mainly through micropinocytosis [90,91]. Internalization of ATP to the cancer cell augments the activity of the drug efflux pathway (i.e., through ABC transporters), which diminishes the intracellular drug concentration and promotes and cancer persistence [89]. In addition, high levels of accumulated intracellular ATP compete with tyrosine kinase inhibitors (TKI) on its receptor (RTK) binding site, which activate phosphorylation and the cascade of cell signaling [92]. Increased ATP internalization promotes TKI translocation (in addition to chemo-drugs) through efflux transporter, which reduces the TKI accumulation inside the cell and increases RTK activity, cell machinery, and resistance [89]. Wang et al. also revealed that drug resistance in the cancer cells mediated by the ability of extracellular ATP molecules to enhance the activity and overexpression of efflux ABC transporters [89]. Shedding light on the blocking/inhibiting mechanisms of extracellular ATP internalization and expression/activity of ABC transporters might substantially affect the chemosensitivity of tumor cells.

## 3. Targets of Natural Products in Cancer Chemo-Resistance

When a specific cancer type exhibits drug resistance to many drugs, this is referred to as the development of multidrug resistance (MDR) [93]. A potential approach to overcome drug resistance is to target the mechanisms of resistance. The general mechanisms of resistance are currently well recognized; they include increased drug efflux and decreased drug influx, drug inactivation, processing of drug-induced damage, alterations in drug target, and evasion of apoptosis. Certain examples of specific mechanism are the expression of resistant transporters or genes that can enhance drug efflux [94]. Drug efflux, facilitated by membrane transport proteins, is associated with the development of MDR in tumor cells [95]. Overexpression of ATP-binding cassette (ABC) membrane transport proteins has been considered the leading contributor to resistance and chemotherapy failure in several types of cancer [96,97,98].

An elevated efflux of chemotherapeutic drugs from cancer cells leads to decreased intracellular drug concentrations by pumping drugs out of cells. Drug efflux transporters are mainly responsible for the development of MDR in cancer cells [7,99]. Membrane transport proteins can eliminate drugs from cells and promote drug redistribution. Drug redistribution reduces drug concentrations in the organelles below lethal concentrations, which further reinforces the drug resistance. Some known proteins related to MDR include P-glycoprotein (P-gp), multidrug resistance protein (MRP), breast cancer resistance protein (BCRP), and lung resistance-related protein (LRP) [7].

### 3.1. P-Glycoprotein

P-glycoprotein (P-gp) is also known as multidrug resistance protein-1 (MDR-1), an ABC sub-family-B member-1 encoded in the human body by the ABCB1 gene [99,100]. ATP binding causes activation of the ATP-binding domains and the hydrolysis of ATP, which will cause change in the transporter shape, essential for the functioning of the transporter and thus results in drug efflux [99]. P-gp has 12 transmembrane domains and two ATP binding sites on its transporter structure that can be inserted into the cell membrane and bind to a variety of chemotherapeutic drugs. P-gp can detect and bind drugs entering the plasma membrane [101,102,103]. Its normal function is to protect cells against xenobiotics and cellular toxicants and thus plays an important role in maintaining physiological homeostasis [104,105]. P-gp expression varies in various types of cancers. Colon, pancreas, liver, adrenal gland, and kidney cancers demonstrate highest levels of P-gp expression, while intermediate P-gp expression is seen in soft tissue carcinomas, neuroblastoma, and hematological malignancies. Breast, ovary, lung, and esophageal cancers initially display low P-gp levels, but the levels of P-gp efflux transporters increase after the cancer shows resistance to the chemotherapeutic treatment [99,101]. P-gp causes decreased intracellular drug concentration, and overexpression of P-gp is always related to MDR [7,101]. Several P-gp inhibitors generations were developed in hope of circumventing MDR, to block P-gp, and to improve the efficacy of chemotherapy in MDR tumors [102,106,107,108]. MDR chemosensitizers are P-gp modulators that administered in combination with cytotoxic agents, which are substrates of the efflux pump could restore their efficacy in resistant cancer cells [109].

First-generation drugs appear less potent, non-selective, and have a low P-gp binding affinity. High doses of these inhibitors are required to reverse MDR which can lead to toxic side effects. Second-generation P-gp inhibitors hinder metabolism and excretion of chemotherapeutic drugs by blocking the effects of P-gp. Some shortcomings of second-generation P-gp inhibitors, such as interaction with cytochrome P450, were overcome in third-generation P-gp inhibitors [99]. Unfortunately, the first three generations have several safety problems, such as unexpected systemic toxicities, non-targeted inhibition, and unpredictable pharmacokinetic interactions between chemotherapeutic agents and candidate P-gp inhibitors. For these reasons alternative strategies are being pursued by scientists to develop a fourth generation of P-gp inhibitors with safety advantages from natural products [107,108,110,111].

Stemofoline, an alkaloid extracted from *Stemona bukilli*, was reported to increase the sensitivity of the chemotherapeutic of MDR leukemic cells and raise the accumulation of P-gp substrates (calcein-AM and rhodamine 123). However, it shows no effect in the P-gp expression according to the Western blot analysis [106]. Moreover, Chang et al. have investigated sesquiterpene pyridine alkaloids (wilforine) and their effect on P-gp expression and function. The study shows that wilforine was able to suppress the efflux activity of P-gp in a concentration-related mode along with re-sensitizing MDR cancer cells to chemotherapy agents [112]. Another study has suggested that tenulin and isotenulin, a natural sesquiterpene lactone, have the potential to be improved for synergistic treatment of MDR cancers. It shows significant prevention of the P-gp activity through triggering P-gp ATPase transporter [104]. Moreover, combination of polyphenols such as EGCG, tannic acid, and curcumin exhibited a high synergistic effect with doxorubicin via attenuating the P-gp function in human colon cancer and leukemia cell lines [102]. Moreover, the Western blot analysis shows a reduction in P-gp levels after applying curcumin treatment in K562/DOX cells as well as enhances the sensitivity of the cells to the chemotherapy [113]. Moreover, the expression of P-gp was decreased in A2780/Taxol cells when curcumin and piperine was combined in solid lipid nanoparticle form [114]. Teng et al. suggested that caffeic acid can reduces cancer MDR in human cervical cells (KB/VIN). It inhibited P-gp efflux via attaching to P-gp through GLU74 and TRY117 residues [103]. Recently, quercetin was also reported to have modulation effect on P-gp expression in HeLa and SiHa cells. According to the Western blot analysis, the co-treatment group (quercetin and cisplatin) showed lower levels of P-gp compared to the single-drug groups [107]. Moreover, other studies have shown a quercetin downregulation effect on P-gp efflux function [108,115,116]. Kaempferol is a natural flavonoid that was able to reverse the multidrug resistance in HepG2and N1S1 liver cancer cells via reducing P-gp overexpression [117]. Emodin is another natural compound that revealed anticancer activity and improved chemotherapy sensitivity in lung cancer (A549 and H460) via reducing P-gp expression [110]. It also reversed drug resistance and enhanced the sensitivity of cisplatin in A549/DDP cells [111]. Ecteinascidin 74, a marine natural product from Caribbean Sea squirts *Ecteinascidia turbinate*, can downregulate P-gp expression at a concentration of 0.1 nM. In addition, it increased the cellular accumulation of DOX/VCR in P-gp-overexpressed cervix cells [118]. Moreover, using a combination treatment of Sophocarpidine from Sophora *flavescens* with vincristine and Adriamycin lowered the expression of P-gp in KBV200 cells [7]. Piperine is an alkaloid found in black pepper (*Piper nigrum*). It has shown downregulation of P-gp, BCRP, MRPs, and ABC transporter genes (ABCB1, ABCG2, and ABCC1), which may reverse MDR in tumor cells [119,120,121,122]. β-Carotene was also reported to modulate P-gp in resistant cancer cell lines (KB-vin and NCI-H460/MX20) and stimulate the basal ATPase activity in a concentration-dependent manner [101,119]. Schisandrin A (Deoxyschizandrin), isolated from Fructus Schizandrae, reversed P-gp-mediated DOX resistance in MCF-7/DOX cells by blocking P-gp, NF-κB, and Stat3 signaling [123]. Moreover, tanshinone microemulsion can significantly reverse drug resistance of K562/ADM cells by inhibiting the P-gp efflux pump effect and increasing the intracellular concentration of chemotherapeutic drugs [124]. Honokiol and magnolol are the main active ingredients in *Magnolia officinali.* They were able to suppress P-gp in NCI/ADR-RES cells and increase the accumulation of P-gp substrate (calcein) in cells. It was found that magnolol can reverse MDR in U937/ADR cells by inhibiting the activity of NF-KB p65 and by downregulating the expression of MDR1 and P-gp [7,125]. Cepharanthine, coumarins, cycloartanes, didehydrostemofolines, eudesmin, and euphocharacins A-L function as P-gp inhibitors in different cancer cell lines [119]. Other phytochemicals that exhibited an inhibition of P-gp are reported in Table 1.

### 3.2. Multidrug Resistance Protein

Multidrug resistance protein (MRP) belongs to the subfamily C in the ABC transporter superfamily of cell membrane transporters known to cause MDR. Multidrug resistance associated protein-1 (MRP-1), encoded in the human body by the ABCC2 gene, has been widely studied for its role in developing drug resistance in various cancers. A distinct feature of MRP1 is that it is a basolateral transporter. This implies that MRP1 activity results in the movement of compounds into cells that lie below the basement membrane. The transporter prevents drug absorption from the basolateral side and clears the drugs out of cells [99]. MRP demonstrated a substrate preference for negatively charged drugs and natural products, such as glutathione, glutathione conjugated leukotrienes, glucosylation, conjugation, sulfation, and glucuronylation [102,103,116,125]. This implies that the mechanism of transport of MRP1 is different from that of P-gp transport [99].

MRP1 is expressed ubiquitously in most of the body, including lung, testis, skeletal, and cardiac muscles. Thus, it is present in most of the tumors, including breast cancer, and plays an important role in MDR. Resistance due to MRP/ABCC members (MRPs 1–3) is often caused by an increased efflux and leads to decreased intracellular accumulation of anticancer drugs. Drug targeting of MRP transporters can help to overcome resistance associated with breast cancer cells [99,100]. The importance of MRPs in cancer therapy is also implied by their clinical insights. Modulating the function of MRPs to re-sensitize chemotherapeutic agents in cancer therapy shows great promise in cancer therapy; thus, multiple MRP inhibitors have been developed recently [126]. Inhibitors of MRP1 are useful to reverse or prevent acquired drug resistance and to sensitize drug-naïve untreated tumors to anticancer drugs [100]. Various natural products exhibit inhibition activity toward MRPs efflux function. Resveratrol, a polyphenol compound, has downregulated p-gp and MRP1 expression in multidrug-resistant human colon cancer (Caco-2) and CE/ADR5000 cells. Moreover, it enhanced doxorubicin cytotoxic activity and increased cell sensitivity to chemotherapy [127]. Moreover, three doxorubicin-resistant cell lines of acute myeloid leukemia were treated with resveratrol, and the results exhibited inhibition of cell growth, a significant reduction in MRP1 expression, and an increased uptake of the MRP1 substrate into the cells [128]. Emodin is a natural compound that belongs to the anthraquinone family [129]. It shows anticancer activity and modulation of chemo-resistance of human bladder cancer cells to cisplatin repressing MRP1 [130]. Recently, Guo et al. demonstrated the effect of emodin on gemcitabine resistance in pancreatic cancer cells. The drug resistance associated proteins have been evaluated in PANC-1 cell xenograft in mice. It was revealed that emodin was able to suppress P-gp, MRP1, and MRP5 expression compared to the control group [131]. On the other hand, treating breast cancer cells resistant to tamoxifen with curcumin caused an enhancement in the sensitivity of cells to the chemotherapy mediated by inhibition of the MDR proteins, particularly MRP2 [132]. Moreover, curcumin was also able to reverse cisplatin chemo-resistance in cervical cancer cells via downregulation of MRP1 and P-gp1expression [133]. Quercetin is a natural polyphenol that has variety of pharmacological activities including the modulation of efflux transporters. It prevented the accumulation of P-gp, BCRP, and MRP2 in triple negative breast cancer cell lines (MDA-MB-231) [134]. Moreover, epigallocatechin-3 gallate (EGCG), a polyphenolic catechin, showed an impact on chemotherapy resistance mediated by the suppression of MDR-related proteins [135,136]. 7,3′,4′-trihydroxyisoflavone (THIF) is the major metabolite of daidzein. It downregulates the MDR1 promoter region and negatively modulates the MDR1 by controlling transcription factors and then generating new MDR. When THIF is combined with adriamycin, the mRNA expression of MRP, MDR1, and MRP2 was lower than that of adriamycin alone [7]. Moreover, strychnine was found to decrease the gene and protein expression of MRP, but not affect the expression of MDR1 [7]. Some other natural products are mentioned in Table 1.

### 3.3. Breast Cancer Resistance Protein

Breast cancer resistance protein (BCRP), encoded in the human body by the ABCG2 gene, was first identified in a drug-resistant human breast cancer cell line. BCRP belongs to the ABCG subfamily of ABC transporters. BCRP is a half-transporter and dimerization is essential to be functional [99,101]. BCRP has one adenosine 5’-triphosphate ABC and six transmembrane domains and is, therefore, a so-called half-ABC transporter; BCRP is likely to form a homodimer or homooligomer in order to obtain functional activity [137,138]. BCRP is mainly expressed in the cell membranes of multiple organs, including the gastrointestinal tract, liver, kidney, brain, endothelium, mammary tissue, testis, and placenta [125,139]. BCRP plays an important role in intercellular drug absorption, metabolism, and excretion, as well as toxicity [99]. BCRP has been extensively studied for its role as an efflux transporter of drugs, leading to drug resistance in target cells and decreased pharmacological effects of substrate drugs. Overexpression of BCRP has been regarded as one of the causes of MDR in different diseases [139]. It causes MDR in most of the types of cancers. In addition to cell membranes, BCRP is also expressed in intracellular vesicles. These vesicles generally retain drugs, but BCRP pumps the drugs out quickly [99]. BCRP actively extrudes a board range of endogenous and exogenous substrates across biological membranes, which include sulfate conjugates, taxanes, carcinogens, glutamated folates, and porphyrins [125]. This is another reason for increased drug resistance due to BCRP efflux transporter. BCRP is highly expressed in side-population cells in breast cancer [99]. A strong correlation between high ABCG2 expression and poor prognosis of leukemia patients has been described [140]. These cells possess stem cell-like properties and are mostly resistant to chemotherapy. BCRP/ABCG2 inhibitors can have additional benefits besides counteracting MDR [99]. Estrogens and antiestrogens have been shown to reverse cancer drug resistance mediated by BCRP [141]. Among many inhibitors, the most promising ones are bivalent flavonoids, which have shown broad-spectrum inhibitory activity as compared to other classes of compounds [99]. Unfortunately, few clinically useful inhibitors of BCRP have been developed. Therefore, a new, specific BCRP inhibitor is still needed to improve outcomes of drug treatment [138,139]. Harmine is a harmala alkaloid that has been used in folk medicine for anticancer therapy [142]. It reversed the resistance of methotrexate and cisplatin drugs in a cancer cell line with BCRP-mediated efflux with no effect on p-gp [132,133,138]. Acacetin, a flavonoid compound, was also reported to have strong reversing activity of BCRP-mediated drug resistance [119,125]. Moreover, apigenin increased the accumulation and inhibition of BCRP in combination with other flavonoids (biochanin A and chrysin) [138,143,144,145]. Biochanin A is an isoflavone found in red clover with antimutagenesis activity. It inhibited MDR-associated proteins including p-gp, MRP1, and BCRP [132,133,138,143]. Other flavonoids such as diosmetin, genistein, kaempferol, luteolin, naringenin-7-glucoside, and quercetin were also reported to have an inhibition effect on BCRP [125]. Recent studies suggested that tangeretin, a natural polymehoxyflavone, showed a potent inhibitory effect on BCRP along with significant suppression of other MDR markers [132,138,143,146]. Terpenoids, hesperetin, rotenoids, stilbenoids, daizein, and other phytochemicals that showed an impact on BCRP arelisted in Table 1.

### 3.4. Lung Resistance Protein

Lung resistance protein (LRP) is another transmembrane protein, which is encoded by the LRP gene [99]. It is also known as major vault protein (MVP or VAULT1); vaults are localized in nuclear pore complexes and are involved in nucleocytoplasmic transport and participate in compound transportation in the nucleoplasm [147,148]. It was first discovered in non-small cell lung cancer cell line SW-1573. The protein is found in the cytoplasm and in the nuclear membrane of tumor cells. It is not a member of the ABC superfamily of transporter proteins [99]. These vaults may play a role in MDR by regulating the nucleo-cytoplasmic movement of drugs. LRP protein is overexpressed in most cancers, which results in lower accumulation of anticancer drugs in the nucleus. LRP also causes resistance to compounds including alkaloids, anthracyclines, and epipodophyllotoxin. In addition to this, LRP also causes resistance to many drugs, which include doxorubicin (Dox), vincristine, cisplatin, and carboplatin [102,113,149]. In contrast to MRP and P-gp, the transmembrane transport region of LRP lacks the ATP-binding site specific to ABC transporters. This region is not associated with the cell membrane but with transport between the nucleus and cytoplasm [124]. Although the function of LRP is still not fully understood, its role in the formation of barrel-shaped vault organelles is recognized. Vaults transport different molecules between nucleus and cytoplasm. LRP is normally expressed in bone marrow. Positive or higher expression has been associated with adverse outcomes in leukemia as well as multiple solid tumors [148]. Many natural products have overcome chemotherapeutic resistance by downregulation of the lung resistance protein. Ginsenoside Rg3 is one of the main ginsenosides derived from ginseng. It effectively prevents tumor cell growth in animal models and cell lines as well as targets the MDR factors in resistant cells such as MDR1, MRP, and LRP [144,150,151]. Moreover, peimine, an alkaloid derived from Fritillaria, was able to reverse the MDR of A549/DDP cell line via suppression of ERCC1 mRNA and LRP expression [152]. Paeonol is another natural compound that mediates the inhibition of LRP, P-gp, MDR1, and MRP in multidrug resistance cells [7]. In gastric cancer patients, the expression level of LRP and MDR1 has been blocked following treatment with Chinese herbal medicine (Shen-qi-jian-wei Tang) [94].

### 3.5. Protein Kinase C

Protein kinase C (PKC) is a phospholipid-dependent, cytoplasmic, serine/threonine kinase with a family composed of at least 12 isozymes [153,154]. These isozymes are classified into three main groups: classical or conventional PKCs (cPKCs; PKCα, PKCβI, PKΧβII and PKCγ), novel PKCs (nPKCs; PKCσ,PKCδ, PKCε, PKCη, and PKCθ), and atypical PKCs (aPKCs; PKCζ, and PKCλ) [155,156]. Moreover, PKC isozymes have various biological activities including receptor desensitization, transcription modulation, immune signaling regulation, cell growth control, as well as learning memory [155]. In terms of cancer biology, PKC isozymes mediate different signal transduction of cell proliferation, differentiation, angiogenesis, and programmed cell death [157,158,159]. Tumorigenesis and drug resistance are associated with the interruption of protein kinase C regulation [153]. Several preclinical studies have shown the effect of blocking PKC on drug resistance and the enhancement of the conventional chemotherapy cytotoxic activity [158,159,160]. Moreover, upregulation of PKC expression in the cytosolic and nuclear compartments was reported in particular MDR tumor cell lines compared to the parent cells [161,162,163,164]. Many phosphorylation reactions and binding of cofactors are controlling the activity of PKC [143]. PKC isozymes may be activated by Ca^+2^, diacylglycerol (DAG), and phospholipids [165]. Moreover, phorbol ester, a tumor promoter, is also able to activate PKCs as it mimics the action of DAG [144]. In MDR cancer cell lines, a study found a correlation between high PKC transduction signaling, particularly cPKC and nPKC, and the upregulation of P-glycoprotein phosphorylation as well as induction of intracellular MDR phenotypes [158,166,167]. Plant-derived products showed great potential to reverse MDR in cancer cells through different mechanisms as inhibiting PKCs is one of these pathways [7]. Curcumin, a polyphenolic compound, was able to suppress the expression of PKC-α and –ζ in breast cancer cell lines (MCF-7 and MDA-MB-231), resulting in sensitizing tumor cells to the chemotherapeutic drugs [145]. Flavonoids such as quercetin also showed an inhibition effect on PKC transduction in hepatocellular carcinoma [146]. Due to the PKC isozyme’s complex role in the cellular functions, inhibition or stimulation of these isoforms might lead to reducing multidrug resistance in cancer cells [168]. Russo et al. found that quercetin mediated CD95-resistant cell line apoptosis via activation of PKCα [169]. The effect of phorbol esters and other diterpenoids on PKCs has been reviewed extensively by Remy and Litaudon (1), demonstrating phorbols interaction with PKCs based on substitution pattern high potency. This interaction involves hydrogen bonding and hydrophobic links, which end up with complex formation with PKCs and reduce their activity [149,150,151]. On the other hand, tigliane diterpenoids are essential derivatives of the Euphorbiaceae family [170], particularly prostratin (a 12-deoxyphorbol ester), which has been reported to be a potent stimulator of PKCs without pro-tumoral activity [157]. Prostratin has potent antiviral and anticancer activities, especially against liver, breast, and gastric tumors [158].

### 3.6. Glutathione Transferase

Glutathione transferases (GSTs) are multifunctional enzymes known as phase II metabolic enzymes that function as cellular detoxifying agents. They can break down the glutathione part of non-polar xenobiotics and endogenous molecules converting them to more water-soluble compounds to ease their removal [159,171,172]. The GST family consists of different isozymes classes including α, Σ, Ζ, Ω μ, π, and θ, which are responsible for catalyzing a wide range of substances [160,173]. Moreover, it was found that a high level of GSTs are associated with developing MDR in cancer cells [61,174,175]. The antioxidant activity of GSTs mediates the chemo-resistance in tumor cells via detoxifying the anticancer drugs and, as a result, reducing cells’ sensitivity to the treatment [173,176]. Several studies have shown the correlation between GST overexpression and chemo-resistance in various types of cancer, such as lung cancer [177,178,179], breast cancer [166,167,180], brain [181,182], and gastric cancer [183,184]. Thus, many synthesized and natural GST inhibitors have been investigated to control the multidrug resistance in cancer cells [159]. Curcumin has known for its antioxidant, anti-inflammatory, and chemopreventive activity [173,185]. It shows an impact on MDR markers by inhibiting GSTπ in the non-small cell lung carcinoma cell line (NCI-H460/R) [186]. Moreover, it reduced drug resistance in melanoma cells by downregulation of GST and MRP1 [187]. Emodin is a type of natural anthraquinone presented in much herbal medicine [188]. It exhibited a reversal effect on multidrug-resistant promyelocytic leukemia (HL-60/ADR cells) and human oral squamous carcinoma (KBV200 cells) via many pathways’ one of them was the reduction of GSTπ [189,190]. Majidinia et al. have also reported the suppression effect of emodin and quercetin on GSTπ to overcome MDR in cancer cells [191]. Moreover, fisetin, a plant flavonol, was significantly able to downregulate GST expression in colorectal adenocarcinoma cells (Caco-2), which made fisetin a promising GST-targeted chemosensitizer for modulating MDR [192]. Yu Ping Feng San (YPFS) is a popular Chinese herbal combination formula composed of Astragali Radix, Atractylodis Macrocephalea Rhizoma, and Saposhnikoviae Radix. Du et al. have tested the activity of (YPFS) on cisplatin-resistant lung cancer (A549/DDP cells). It reduced MDR-associated proteins and enzymes such as ATP-binding cassette transporters and GSTs isozymes [178]. Another mixture of Chinese supplement energy and nourish lung (SENL) herbs consisting of ginsenoside Rg1, ginsenoside Rb1, ginsenoside Rg3, astragaloside IV, ophiopogonin D, and tetrandrine has been investigated in A549/DDP cells. It reduced the GSTπ expression and reversed the cisplatin resistance in lung cancer xenografts [193]. Furthermore, ginger phytochemicals (6-gingerol, 10-gingerol, 4-shogaol, 6-shogaol, 10-shogaol, and 6-dehydrogingerdione) inhibited GSTπ and MRP1 in docetaxel-resistant prostate cancer (PC3R) [194]. Another study suggested that oridonin, a tetracyclic diterpenoid extracted from *Rabdosia labtea*, stimulated the apoptosis-associated markers in gemcitabine-resistant PANC-1 pancreatic cancer cells. It suppressed the expression of GSTπ and lipoprotein receptor protein 1 (LRP1) [195]. Natural phenols such as resveratrol have shown modulation of multidrug resistance in tumor cells. Treating doxorubicin-resistant Caco-2 cells with resveratrol revealed a significant reduction in GST mRNA levels along with various MDR markers [127]. Moreover, dietary carotenoids particularly fucoxanthin (FUC), a non-pro-vitamin A carotenoid found in brown seaweeds, have displayed antioxidant potential and improved many cancer cells’ sensitivity toward chemotherapies [196,197]. Eid et al. demonstrated the effect of FUC on enhancing doxorubicin activity and mediated apoptosis via increasing caspases and p53 as well as downregulation of GST, CYP3A4, and PXR in resistant cancer cells [166].

### 3.7. Topoisomerases

DNA topoisomerases (topo) are enzymes found in the nucleus of cells. They regulate DNA replication, repair, and chromosomal segregation by converting DNA topology [198]. There are two kinds of topoisomerases: topo I and II, with different classes implementing various functions. Topo I catalyzes the breaking of single strands of DNA, while topo II cutting the double strands of DNA to relieve the supercoiling [199,200]. Cell-cycle arrest and cell death by apoptosis are the results of blocking one type of topoisomerase, while blocking the two types can highly improve the cytotoxicity toward cancer cells [201,202]. Many cancer cells have shown a high level of topo II expression, which makes it a target for new chemotherapy [203]. Topo II has two main isoforms: topo IIα and topo IIβ [204,205]. Since topo IIα has an important role in cell growth, it is highly expressed in fast-growing cancer cells. On the other hand, topo IIβ is present in dormant cells in all kinds of tissues during the whole cell cycle [205,206]. Many powerful chemotherapy drugs such as doxorubicin, teniposide, and etoposide are topoisomerase II inhibitors [205]. However, serious side effects could result from using these drugs due to the lack of selectivity as well as the risk of drug resistance due to the enzymes’ gene mutation or dysregulation of their expression in tumor cells [194,207,208,209]. Thus, looking for new phytochemicals that targeting topoisomerases enzyme is a promising branch in chemotherapy development. Many secondary metabolites have an impact on topoisomerase enzymes such as alkaloids, flavonoids, and triterpenes [201,210,211,212,213]. Emodin is an example of a natural product that reversed the multidrug resistance in promyelocytic leukemia (HL-60/ADR cells). It reduced the expression of MDR proteins including topo IIβ and MRP1 along with increasing the intracellular accumulation of adriamycin (ADR) and daunorubicin (DNR) [189]. This effect was also reported in resistant human oral squamous carcinoma cells [190]. Moreover, curcumin was able to downregulate the topo IIα in human non-small cell lung carcinoma cells (NCI-H460/R cells) [186]. Riccardin D is a macrocyclic bisbibenzyl extracted from the Chinese liverwort plant. It promoted apoptosis and reduce MDR in leukemia cells via inhibition of topoisomerase II and decreasing p-gp expression [214].

### 3.8. Hypoxia-Inducible Factor

Hypoxia usually developed in rapidly growing cancer cells. It is a major problem in achieving effective cancer chemotherapy [215,216]. Tumor hypoxia has been known to stimulate the expression of several genes correlated with drug resistance [217]. Hypoxia-inducible factor-1 is an oxygen-sensitive heterodimeric transcription factor composed of two subunits: α and β [217,218,219]. It was reported that chemoresistance is associated with a high level of HIF-1α expression in many cancer types including ovarian cancer, hepatocellular carcinoma, glioblastoma, and colorectal cancer [220,221,222,223]. Moreover, HIF-1α can trigger more than 60 genes involved in tumor growth, metastasis, cellular metabolism, the reduction of apoptosis, and poor prognosis [224,225]. HIF-1α follows different pathways to promote tumor drug resistance, and one of them is by regulating MDR-associated proteins such as p-gp and MRP1 [226,227]. Natural products and their derivatives are an abundant source of safe and effective resistance reversal agents [228]. Epigallocatechin-3-gallate (EGCG) is a polyphenol extracted from green tea and one of the MDR reversal modulators [228,229]. Wen et al. suggested that an EGCG derivative reduced drug resistance in doxorubicin-resistant human hepatocellular carcinoma cells (BEL-7404/DOX) via downregulation of HIF-1α and p-gp [228]. Moreover, apigenin, a type of flavonoid, reversed paclitaxel resistance in hypoxic-liver tumor cells by inhibiting HIF-1α [230]. Treating drug-resistant prostate carcinoma cells (DU-145 cell line) with emodin improved the efficacy of cisplatin and attenuated MDR markers expression and suppressed HIF-1α [207,231]. Interestingly, quercetin was able to inhibit HIF-1α and MDR1 as a result of the enhanced cytotoxic activity of doxorubicin and gemcitabine in pancreatic and liver cancer cells [232]. On the other hand, resveratrol repressed the hypoxia-induced resistance to doxorubicin in MCF-7 cells via downregulation of HIF-1α protein expression [208]. Nuciferine is an aromatic alkaloid extracted from lotus leaves that exhibited anti-inflammatory, antioxidant, and anticancer properties [209,233,234]. Recently, nuciferine has been applied in drug-resistant tumor cells, and it was able to regulate MDR proteins as well as reduce the activation of Nrf2 and HIF-1α [235]. Modulation of HIF-1α by curcumin was also reported [236,237]. Figure 2 displays the main natural products and their targets in cancer multidrug resistance.

**Table 1 biomedicines-09-01353-t001:** Natural products with their mechanisms of inhibition.

Substance	Mechanism of Inhibition	References
Dauriporphine	↓ P-g expression	[125]
Glaucine	↓ P-g expression ↓ MDR1 ↓ MRP1	[125]
Hernandezine	↓ P-g expression	[125]
Antofine	↓ P-g expression ↓ MDR1 mRNA	[125]
Harmine	↓ BCRP	[132,133,138]
Tryptanthrin	↓ P-g expression ↓ MRP2	[125]
Lobeline (from *Lobelia inflate*)	↓ P-g expression	[120,125]
Tetramethylpyrazine	↓ P-g expression ↓ MDR1 mRNA ↓ MRP1, MRP2, MRP3	[105,138,144,238]
Danshensu and tetramethylpyrazine (from the Chinese herbs)	↓ P-g expression	[239]
Acrimarine E	↓ P-g expression	[125]
Gravacridonetriol	↓ MDR1 mRNA	[125]
2-Methoxycitpressine I	↓ P-g expression	[125]
Capsaicin (extracted from *Capsicum annuum*)	↓ P-g expression	[100,125]
Acacetin	↓ BCRP ↓ MRP1	[119,125]
Amorphigenin	↓ P-g expression	[125]
Apigenin	↓ BCRP ↓ MRP1 ↓ P-g expression ↓ HIF-1α	[7,138,143,144,240]
Ampelopsin	↓ P-g expression	[125]
Biochanin A	↓ BCRP ↓ MRP1 ↓ P-g expression	[132,133,138,143]
Catechin	↓ ATPase activity ↓ P-g expression	[125]
Chalcone	↓ MRP1 ↓ P-g expression	[125] [141]
Chrysin	↓ BCRP ↓ P-g expression	[125] [119,141]
Diosmetin	↓ BCRP	[125]
Green tea catechins (EGCG, ECG, CG)	↓ P-g expression ↓ MDR1 ↓ ATPase activity	[100,141]
Epicatechin gallate	↓ P-g expression	[118,125]
Epigallocatechin gallate	↓ P-g expression ↓ MDR1 ↓ ABCG2 ↓ HIF-1α	[113,114,132,138,143,144,241]
Formononetin	↓ P-g expression	[125]
Genistein	↓ BCRP ↓ MRP1	[132,133,138,143]
Glabridin	↓ P-g expression	[7,114,138]
3,3’,4’,5,6,7,8- Heptamethoxyflavone	↓ P-g expression	[125]
Kaempferol	↓ BCRP ↓ MRP1 ↓ P-g expression	[7,134,138,143]
Luteolin	↓ BCRP ↓ MRP1	[125,141]
Morin	↓ P-g expression ↓ MRP1	[132,133,138,143]
Myricetin	↓ MRP1 and MRP2 activity ↓ Calcein efflux	[7,105,133,138,143]
Naringenin	↓ P-g expression	[134,138,143]
Naringenin-7-glucosid	↓ BCRP	[125]
Nobiletin (found in citrus fruit)	↓ P-g expression ↓ MRP1	[114,138,143,146]
Phloretin	↓ P-g expression ↓ MRP1	[132,133,138,143]
Procyanidine	↓ P-g expression	[125]
Quercetin	↓ MRP1-mediated drug transport ↓ BCRP ↓ MRP1, 4 and 5. ↓ P-g expression ↓ PKC ↓ HIF-1α ↓ MDR1	[7,118,119,120,124,125,141,146,169,232,242,243,244,245,246,247]
Robinetin	↓ MRP1 and MRP2 activity (inhibited calcein efflux)	[125]
Rotenone	↓ P-g expression	[125]
Silymarin	↓ P-gp ATPase activity ↓ P-gp-mediated cellular efflux ↓ [3 H]azidopine photoaffinity labeling of P-gp suggesting a direct interaction with the P-gp substrate binding ↓ MRP1-mediated drug transport ↓ BCRP	[7,132,133,138,143]
Tangeretin	↓ P-g expression ↓ BCRP	[132,138,143,146]
Curcumin	↓ P-g expression ↓ BCRP ↓ MRP1 ↓ MDR1 mRNA ↓ ABCG2 and ABCC1 ↓ PKC-α and –ζ ↓ GSTπ ↓ Topo IIα ↓ HIF-1α	[7,94,100,118,119,120,124,125,145,186,236,237,242,246,248,249,250,251]
Matairesinol (found in soybean (Glycine max))	↓ P-g expression ↓ MRP1	[100,125]
Sesamin	↓ P-g expression	[100,125]
Gomisin A	↓ P-g expression	[125]
Schisandrol A	↓ P-g expression	[125] [119]
Chlorogenic acid	↓ P-gp ATPase activity	[125]
Ginkgolic acid	↑ DNR accumulation ↓ P-g expression	[125]
Agnuside	↓ P-gp ATPase activity	[125]
Picroside-II	↓ P-gp ATPase activity	[125]
Santonin	↓ P-gp ATPase activity	[125]
beta-Amyrin	↓ P-g expression	[125]
Glycyrrhetinic acid (Enoxolone) (Licorice)	↓ P-g expression ↓ MRP1	[100,125]
Obacunone	↓ P-g expression	[125]
Oleanolic acid	↓ P-g expression	[125,247]
Uvaol	↓ P-g expression	[125,247]
Alisol B 23-acetate	↓ P-g expression	[113,133,138]
Ginsenoside Rg_3_	↓ Binding of [3 H] azidopine to P-gp ↓ P-g expression	[119,125]
Protopanaxatriol ginsenosides 20S-ginsenoside Ginsenoside Rb1 Ginsenoside Rg3	↓ P-g expression ↓ BCRP ↓ MRP1 ↓ MDR1 ↓ LRP	[113,133,138,144,146]
Tenacigenin B: P8, P26 and P27	↓ P-g expression ↓ MRP1 ↓ ABCG2-mediated efflux	[125]
Tenacigenin B: P2, P3 and P6	↓ P-g expression ↓ MRP1	[125]
Tenacigenin B: P1, P4, P5, P9 and P28	↓ P-g expression	[125]
Aurochrome	↓ P-g expression	[125]
Diepoxycarotene	↓ P-g expression	[125]
Mutatochrome	↓ P-g expression	[125]
Clausarin	↓ P-gp-mediated drug efflux	[125]
Phyllodulcin	↑ DNR accumulation (inhibition of P-gp-mediated efflux of DNR)	[125]
Acteoside (Verbascosine)	↓ P-gp ATPase activity	[125]
Berbamine	↓ MDR1 gene expression	[125,242]
Glaucine	↓ P-g expression ↓ MRP1 ↓ MDR1 and MRP1 genes	[125]
Fangchinoline	↓ P-g expression	[125]
O-(4-ethoxyl-butyl)- berbamine	↓ MDR1 gene expression	[125]
Tetrandrine (dried root of *Stephania tetrandra*)	↓ P-g expression ↓ LRP	[103,113,135,138,144]
Matrine	↓ P-g expression	[125]
Antofine	↓ MDR1 mRNA ↓ P-g expression	[125]
Ephedrine	↓ MDR1 mRNA ↓ P-g expression	[125,242]
Indole-3-carbinol	↓ P-g expression	[125]
Staurosporine	↓ P-g expression ↓ MDR1 gene expression	[125]
Vauqueline	↓ MDR1 mRNA ↓ P-g expression	[125]
Gravacridonetriol	↓ MDR1 mRNA	[125]
Clitocine	↓ MDR1 mRNA ↓ P-g expression)	[125]
Sulfinosine	↓ MDR1 mRNA ↓ P-g expression	[125]
Bisdemethoxycurcumin	↓ P-gp expression ↓ MDR1	[118,125]
Honokiol and magnolol (isolated from *Magnolia officinali*)	↓ MDR1 ↓ P-gp expression	[7,125]
Schisandrin A (Deoxyschizandrin)	↓ P-gp expression ↓ MDR1 ↓ PKC	[104,113,125,133,137,146]
Schisandrin B (Sch B)	↓ P-gp expression and P-gp mediated efflux of Dox. ↓ MRP1	[93]
Triptolide	↓ MDR1 ↓ MRP1 protein expression	[125]
Pyranocoumarins	↓ P-gp expression ↓ MDR1 mRNA expression	[119,125]
Ginger phytochemicals (6-Gingerol,10- Gingerol)	↓ P-gp expression ↓ MRP1	[100,125]
Ginger phytochemicals (6-gingerol, 10-gingerol, 4-shogaol, 6-shogaol, 10-shogaol, and 6-dehydrogingerdione)	↓ GSTπ ↓ MRP1	[194]
*Alisma orientalis*	↓ P-gp expression	[250]
*Piper methysticum*	↓ P-gp expression	[250]
Guggulsterone	↓ P-gp expression ↓ MRPs	[113,114,134,252]
Phenolic diterpenes	↓ P-gp expression	[250]
Vincristine	↓ P-gp expression	[250]
5-Bromotetrandrine	↓ P-gp expression	[119]
Abietane diterpene	↓ P-gp expression	[119]
Amooranin	↓ P-gp expression	[119]
Baicalein and derivatives	↓ P-gp expression ↓ MRPs	[118,119,120,124,141,247]
Bitter melon extract	↓ P-gp expression	[119]
Bufalin	↓ P-gp expression	[119]
Cannabinoids	↓ P-gp expression ↓ BCRP ↓ MRPs	[119]
β-Carotene	↓ P-gp expression	[101,119]
Fucoxanthin	↓ GST	[166]
Catechins	↓ P-gp expression	[111,133,143]
Cepharanthine	↓ P-gp expression ↓ MRP1	[119]
Coumarins	↓ P-gp expression	[119]
Cycloartanes	↓ P-gp expression	[119]
Didehydrostemofolines	↓ P-gp expression	[119]
Eudesmin	↓ P-gp expression	[119]
Euphocharacins A-L	↓ P-gp expression	[119]
Ginkgo biloba extract	↓ P-gp expression ↓ MRP1	[119]
Grapefruit juice extracts	↓ P-gp expression	[119]
Hapalosin	↓ P-gp expression	[119]
Hypericin and hyperforin	↓ P-gp expression ↓ BCRP	[119,246]
Isoquinoline alkaloid, isotetrandrine	↓ P-gp expression	[119]
Isostemofoline	↓ P-gp expression	[119]
Jatrophanes	↓ P-gp expression	[119]
*Kaempferia parviflora* extracts	↓ P-gp expression ↓ MRP1	[119]
Kavalactones	↓ P-gp expression	[119]
Ningalin B and derivatives	↓ P-gp expression	[119]
Opiates	↓ P-gp expression	[119]
Piperine	↓ P-gp expression ↓ BCRP ↓ MRPs ↓ ABC transporter genes (ABCB1, ABCG2, and ABCC1)	[119,120,121,122]
Polyoxypregnanes	↓ P-gp expression	[119]
Sesquiterpenes	↓ P-gp expression	[119,247]
Tenulin	↓ P-gp expression	[104]
Sinensetin	↓ P-gp expression	[119,247]
Taxane derivatives	↓ P-gp expression	[119]
Terpenoids	↓ P-gp expression ↓ BCRP	[119] [246]
Tetrandine	↓ P-gp expression	[119]
Vitamin E TPGS	↓ P-gp expression	[119]
3′-4′-7-Trimethoxyflavone	↓ BCRP	[119,141]
6-Prenylchrysin	↓ BCRP	[119,141]
Eupatin	↓ BCRP	[119]
Daizein	↓ BCRP	[119]
Hesperetin	↓ BCRP	[119,141,244]
Plumbagin	↓ BCRP	[119]
Resveratrol	↓ BCRP ↓ P-gp expression ↓ HIF-1α ↓ GST mRNA expression	[114,133,140,253]
Rotenoids	↓ BCRP	[119]
Stilbenoids	↓ BCRP	[119]
Tectochrysin	↓ BCRP	[119,141]
Tetrahydrocurcumin	↓ BCRP	[119]
Ligustrazine	↓ Expression of P-gp	[7]
Sophocarpidine	↓ Expression of P-gp	[7]
Strychnine	↓ Gene and protein expression of MRP	[7]
Three hydroxyl soy isoflavone	↓ MRP, MDR1, MRP2	[7]
Ecteinascidin	↓ P-gp expression	[7]
Ecteinascidin 743	↓ P-gp expression ↑ Cellular accumulation of DOX/VCR in P-gp-overexpressed cervix cells	[118]
7,3′,4′-trihydroxyisoflavone	↓ mRNA expression of MRP, MDR1, and MRP2	[7]
Paeonol (extracted from the dry velamen of *peony* or any part of *Cynanchum paniculatum*)	↓ P-g expression ↓ MDR1 ↓ MRP ↓ LRP	[7]
Oroxylin A-7-glucuronide	↓ MDR1 ↓ P-g expression	[7]
3′,4′,5′,5,7-pentamethoxyflavone (PMF) and derivatives of *epimedium*	↓ MDR1 ↓ P-g expression	[7]
Osthole (isolated from *Fructus Cnidii*)	↓ P-g expression	[7]
Praeruptorin A (extracted from *Radix Peucedani*)	↓MDR1 and P-gp mRNA	[7,247]
Diphyllin	↓ P-gp expression	[7]
Emodin	↓ P-gp expression ↓ MRP1 ↓ GSTπ ↓ Topo IIβ ↓ HIF-1α	[113,144,215,216,254,255,256]
Psoralen	↓ P-g expression	[7,242]
Gypenoside	↓ BCRP ↓ P-gp expression ↓ MRP1	[7]
Allicin	↓ MDR1 ↓ P-g expression	[7]
Taccalonolide A and B (extracted from *Tacca chantrieri*)	↓ P-g expression	[7]
Oridonin	↓ P-gp expression ↓ GSTπ ↓ LRP1	[113,144,222]
Ursolic acid (found in *Rosmarinus officinalis*)	↓ P-gp expression	[7,100]
Sipholenol A (found in *sponge Callyspongia siphonella*)	↓ P-g expression	[113,132,146]
Cantharidin (extracted from *Mylabris phalerata Pallas* or *Mylabris cichorii* L.)	↓ P-g expression	[7]
Beta-Elemene (isolated from *Aeruginous Turmeric rhizome*)	↓ P-g expression ↓ MRP	[7] [242]
As2O3, or white arsenic Arsenic Trioxide	↓ P-gp expression ↓ MRP	[7,242]
Artemisinin	↓ P-gp expression	[242]
Artesunate	↓ P-gp expression	[242]
Baicalin	↓ P-gp expression ↓ MRP1	[242]
Berberine (isolated from ancient Chinese herb *Coptis chinensis* French)	↓ P-gp expression ↓ABCG2	[242] [246]
Carnosic acid (Rosemary)	↓ P-gp expression	[114,134,144]
Chelerythrine	↓ P-gp expression	[242]
Gambogic acid	↓ P-gp expression	[242]
Neferine	↓ P-gp expression	[242]
Oxymatrine	↓ P-gp expression	[242]
Peimine	↓ LRP	[242]
Sodium norcantharidate	↓ P-gp expression ↓ MRP	[242]
*Brucea Javanica*	↓ P-gp expression ↓ MRP	[242]
Cinobufacini	↓ P-gp expression ↓ MRP1	[242]
Grape seed polyphenols	↓ P-gp expression	[124,242]
Hyaluronate Oligomers	↓ P-gp expression ↓ MRP	[242]
Jew ear	↓ P-gp expression ↓ MRP	[242]
*Radix notoginseng*	↓ P-gp expression	[242]
*Rhizoma pinelliae*	↓ P-gp expression	[242]
Realgar	↓ P-gp expression	[242]
*Thallus laminariae*	↓ P-gp expression	[242]
Algerian propolis	↓ transport function of P-gp-pump	[257]
Dihydroptychantol A (isolated from *A. angusta*)	↓ P-g expression	[250] [258]
Riccardin F (isolated from *P. intermedium*)	↓ P-gp expression	[258]
Riccardin D	↓ Topo II ↓ P-gp expression	[214]
Andrographolid	↓ P-gp expression	[94]
Parthenolide	↓ Pgp expression	[94]
Rhei Rhizoma, Scutellariae Radix, Poria, Zizyphi Fructus, Zingiberis Rhizoma, Asiasari Radix, Sophorae Radix (herbal extract)	↓ P-gp expression	[94]
Tripterygium wilfordii	↓ P-gp expression ↓ EGFR	[94]
Shenghe Powder (consisting of Radix codonopsis pilosulae, Radix pseudostellariae, Radix scrophulariae, Rhizoma atractylodis macrocephalae, and 6 additional herbs)	↓ P-gp expression	[94]
Shen-qi-jian-wei Tang	↓ MDR1 ↓ LRP	[94]
Yu Ping Feng San (YPFS) (Astragali Radix, Atractylodis Macrocephalea Rhizoma, and Saposhnikoviae Radix)	↓ ATP-binding cassette transporters ↓ GST	[178]
Chinese supplement energy and nourish lung (SENL) herbs (ginsenoside Rg1, ginsenoside Rb1, ginsenoside Rg3, astragaloside IV, ophiopogonin D, and tetrandrine)	↓ GSTπ	[193]
Icaritin	↓ P-gp expression	[118,120]
Icariin	↓ P-gp expression	[118]
Sesquiterpene ester 1	↓ P-gp expression	[118]
Celafolin A-1	↓ P-gp expression	[118]
Celorbicol ester	↓ P-gp expression	[118]
Demethoxycurcumin	↓ P-gp expression	[118]
Euphomelliferine	↓ P-gp expression	[118]
Euphodendroidin D	↓ P-gp expression	[118,247]
Pepluanin A	↓ P-gp expression	[118,247]
Sipholenone E	↓ P-gp expression	[118,247]
Siphonellinol D	↓ P-gp expression	[118]
GUT-70 (From *C. Brasiliense*)	↓ P-gp expression	[118]
Lamellarin I	↓ P-gp expression	[118]
Wogonin	↓ P-gp expression ↓ MRP1	[118]
Aposterol A	↓ P-gp expression ↓ MRP1	[118]
Fumitremorgin C	↓ BCRP ↓ P-gp expression ↓ MRP1	[109,132,162]
Tryprostatin A	↓ BCRP	[118]
Terrein	↓ BCRP	[118]
Lamellarin O	↓ BCRP ↓ P-gp expression	[118]
Secalonic acid D	↓ BCRP ↓ P-gp expression ↓ MRP1	[118]
Quinine and its isomer quinidine	↓ P-gp expression	[120]
Reserpine and yohimbine (isolated from *Rauwolfia serpentine*)	↓ BCRP ↓ P-gp expression	[120]
Bromocriptine ergot alkaloid	↓ P-gp expression	[120]
β-Sitosterol-O-glucoside	↓ P-gp expression	[120]
cardiotonic steroid 3	↓ P-gp expression	[120]
Menthol	↓ P-gp expression	[120]
Aromadendrene	↓ P-gp expression	[120]
Citronellal	↓ P-gp expression	[120]
Citronellol	↓ P-gp expression	[120]
Carnosol	↓ P-gp expression	[100,120]
Limonin	↓ P-gp expression	[120]
Kaempferide	↓ BCRP ↓ P-gp expression	[141,244]
Diosmin	↓ P-gp expression	[244]
Daidzein	↓ BCRP	[141,244]
Tanshinone microemulsion	↓ P-gp expression	[124]
Tea polyphenol	↓ P-gp expression	[124]
Stemocurtisine	↓ P-gp expression	[120]
Stemofoline	↓ P-gp expression	[120]
Oxystemokerrine	↓ P-gp expression	[120]
Amurensin G (from Vitis amurensis)	↓ P-gp expression	[241]
Sakuranetin	↓ P-gp expression	[141]
Floretin	↓ P-gp expression	[141]
Fisetin	↓ P-gp expression ↓ GST	[141,192]
Xanthohumol (derived from *Humulus lupulus*)	↓ mRNA expression of P-gp, MRP1, MRP2 and MRP3	[141]
Silybin (isolated from *Silybum marianum*)	↓ MRP1 ↓ P-gp expression	[141]
Sophoraisoflavone A	↓ MRP1	[141]
LANGDU (a traditional herbal medicine)	↓ P-g expression	[246]
Tanshinone IIA (isolated from *Salvia miltiorrhiza*)	↓ MRP1 ↓ BCRP ↓ P-g expression	[246]
Auraptene (grapefruit)	↓ P-g expression	[100]
Nimbolide	↓ P-gp gene	[140]
*Marsdenia tenacissima*	↓ P-g expression ↓ ABCG2 ↓ MRP1	[97]
Taxifolin	↓ ABCB1 ↓ P-gp expression	[243]
*Heterotheca inuloides* Cass.	↓ MDR1 ↓ MRP1 ↓ BCRP	[98]
Saikosaponin D	↓ MDR1 gene ↓ P-gp expression	[259]
Kanglaite (isolated from *Coix lacryma-jobi*)	↓ MDR1 ↓ MRP2 ↓ BCRP	[240]
*Astragalus membranaceus* polysaccharides Astragaloside II, another component from *A. membranaceus*	↓ P-gp expression ↓ MDR1	[254]
Wilforine	↓ P-gp expression	[112]
*Boswellia serrata* extracts 3- O-acetyl-11-keto-β-boswellic acid (AKBA), the major active ingredient of the gum resin from Boswellia serrata and Boswellia carteri Birdw	↓ P-gp expression	[255]
Pervilleine F	↓ P-gp expression	[247]
Ellipticine	↓ P-gp expression	[247]
Cnidiadin	↓ P-gp expression	[247]
Conferone	↓ P-gp expression	[247]
Rivulobirin A	↓ P-gp expression	[247]
Dicamphanoyl khellactone (*DCK*)	↓ P-gp expression	[247]
Cannabidiol	↓ P-gp expression	[247]
Taccalonolides A	↓ P-gp expression	[247]
Jolkinol B	↓ P-gp expression	[247]
Portlanquinol	↓ P-gp expression	[247]
Dihydro-β-agarofuran	↓ P-gp expression	[247]
Pentadeca-(8,13)-dien-11-yn-2-one	↓ P-gp expression	[247]
Silibinin	↓ P-gp expression	[247]
Nirtetralin	↓ P-gp expression	[247]
Cordycepin	↓ P-gp expression	[104]
Nuciferine	↓ HIF-1α	[235]
Dauriporphine	↓ P-g expression	[138]
Glaucine	↓ P-g expression ↓ MDR1 ↓ MRP1	[138]
Hernandezine	↓ P-g expression	[138]
Antofine	↓ P-g expression ↓ MDR1 mRNA	[138]
Harmine	↓ BCRP	[132,133,138]
Tryptanthrin	↓ P-g expression ↓ MRP2	[138]
Lobeline (from *Lobelia inflate*)	↓ P-g expression	[134,138]
Tetramethylpyrazine	↓ P-g expression ↓ MDR1 mRNA ↓ MRP1, MRP2, MRP3	[105,138,144,238]
Danshensu and tetramethylpyrazine (from the Chinese herbs)	↓ P-g expression	[238]
Acrimarine E	↓ P-g expression	[138]
Gravacridonetriol	↓ MDR1 mRNA	[138]
2-Methoxycitpressine I	↓ P-g expression	[138]
Capsaicin (extracted from *Capsicum annuum*)	↓ P-g expression	[114,138]
Acacetin	↓ BCRP ↓ MRP1	[133,138]
Amorphigenin	↓ P-g expression	[138]
Apigenin	↓ BCRP ↓ MRP1 ↓ P-g expression ↓ HIF-1α	[138,143,144,145,240]
Ampelopsin	↓ P-g expression	[138]
Biochanin A	↓ BCRP ↓ MRP1 ↓ P-g expression	[132,133,138,143]
Catechin	↓ ATPase activity ↓ P-g expression	[138]
Chalcone	↓ MRP1 ↓ P-g expression	[138] [143]
Chrysin	↓ BCRP ↓ P-g expression	[138] [133,143]
Diosmetin	↓ BCRP	[138]
Green tea catechins (EGCG, ECG, CG)	↓ P-g expression ↓ MDR1 ↓ ATPase activity	[114,143]
Epicatechin gallate	↓ P-g expression	[132,138]
Epigallocatechin gallate	↓ P-g expression ↓ MDR1 ↓ ABCG2 ↓ HIF-1α	[113,114,132,138,143,144,241]
Formononetin	↓ P-g expression	[138]
Genistein	↓ BCRP ↓ MRP1	[132,133,138,143]
Glabridin	↓ P-g expression	[114,138,145]
3,3′,4′,5,6,7,8- Heptamethoxyflavone	↓ P-g expression	[138]
Kaempferol	↓ BCRP ↓ MRP1 ↓ P-g expression	[134,138,143,145]
Luteolin	↓ BCRP ↓ MRP1	[138,143]
Morin	↓ P-g expression ↓ MRP1	[132,133,138,143]
Myricetin	↓ MRP1 and MRP2 activity ↓ Calcein efflux	[105,133,138,143,145]
Naringenin	↓ P-g expression	[134,138,143]
Naringenin-7-glucosid	↓ BCRP	[138]
Nobiletin (found in citrus fruit)	↓ P-g expression ↓ MRP1	[114,138,143,146]
Phloretin	↓ P-g expression ↓ MRP1	[132,133,138,143]
Procyanidine	↓ P-g expression	[138]
Quercetin	↓ MRP1-mediated drug transport ↓ BCRP ↓ MRP1, 4 and 5. ↓ P-g expression ↓ PKC ↓ HIF-1α ↓ MDR1	[103,111,113,132,133,134,138,143,144,145,146,181,182,260,261,262]
Robinetin	↓ MRP1 and MRP2 activity (inhibited calcein efflux)	[138]
Rotenone	↓ P-g expression	[138]
Silymarin	↓ P-gp ATPase activity ↓ P-gp-mediated cellular efflux ↓ [3 H]azidopine photoaffinity labeling of P-gp suggesting a direct interaction with the P-gp substrate binding ↓ MRP1-mediated drug transport ↓ BCRP	[132,133,138,143,145]
Tangeretin	↓ P-g expression ↓ BCRP	[132,138,143,146]
Curcumin	↓ P-g expression ↓ BCRP ↓ MRP1 ↓ MDR1 mRNA ↓ ABCG2 and ABCC1 ↓ PKC-α and –ζ ↓ GSTπ ↓ Topo IIα ↓ HIF-1α	[103,105,113,114,132,133,134,138,144,180,212,252,262,263,264,265,266,267]
Matairesinol (found in soybean (Glycine max))	↓ P-g expression ↓ MRP1	[114,138]
Sesamin	↓ P-g expression	[114,138]
Gomisin A	↓ P-g expression	[138]
Schisandrol A	↓ P-g expression	[138] [133]
Chlorogenic acid	↓ P-gp ATPase activity	[138]
Ginkgolic acid	↑ DNR accumulation ↓ P-g expression	[138]
Agnuside	↓ P-gp ATPase activity	[138]
Picroside-II	↓ P-gp ATPase activity	[138]
Santonin	↓ P-gp ATPase activity	[138]
beta-Amyrin	↓ P-g expression	[138]
Glycyrrhetinic acid (Enoxolone) (Licorice)	↓ P-g expression ↓ MRP1	[114,138]
Obacunone	↓ P-g expression	[138]
Oleanolic acid	↓ P-g expression	[138,146]
Uvaol	↓ P-g expression	[138,146]
Alisol B 23-acetate	↓ P-g expression	[113,133,138]
Ginsenoside Rg_3_	↓ Binding of [3 H] azidopine to P-gp ↓ P-g expression	[133,138]
Protopanaxatriol ginsenosides 20S-ginsenoside Ginsenoside Rb1 Ginsenoside Rg3	↓ P-g expression ↓ BCRP ↓ MRP1 ↓ MDR1 ↓ LRP	[113,133,138,144,146]
Tenacigenin B: P8, P26 and P27	↓ P-g expression ↓ MRP1 ↓ ABCG2-mediated efflux	[138]
Tenacigenin B: P2, P3 and P6	↓ P-g expression ↓ MRP1	[138]
Tenacigenin B: P1, P4, P5, P9 and P28	↓ P-g expression	[138]
Aurochrome	↓ P-g expression	[138]
Diepoxycarotene	↓ P-g expression	[138]
Mutatochrome	↓ P-g expression	[138]
Clausarin	↓ P-gp-mediated drug efflux	[138]
Phyllodulcin	↑ DNR accumulation (inhibition of P-gp-mediated efflux of DNR)	[138]
Acteoside (Verbascosine)	↓ P-gp ATPase activity	[138]
Berbamine	↓ MDR1 gene expression	[138,144]
Glaucine	↓ P-g expression ↓ MRP1 ↓ MDR1 and MRP1 genes	[138]
Fangchinoline	↓ P-g expression	[138]
O-(4-ethoxyl-butyl)- berbamine	↓ MDR1 gene expression	[138]
Tetrandrine (dried root of *Stephania tetrandra*)	↓ P-g expression ↓ LRP	[103,113,135,138,144]
Matrine	↓ P-g expression	[138]
Antofine	↓ MDR1 mRNA ↓ P-g expression	[138]
Ephedrine	↓ MDR1 mRNA ↓ P-g expression	[138,144]
Indole-3-carbinol	↓ P-g expression	[138]
Staurosporine	↓ P-g expression ↓ MDR1 gene expression	[138]
Vauqueline	↓ MDR1 mRNA ↓ P-g expression	[138]
Gravacridonetriol	↓ MDR1 mRNA	[138]
Clitocine	↓ MDR1 mRNA ↓ P-g expression)	[138]
Sulfinosine	↓ MDR1 mRNA ↓ P-g expression	[138]
Bisdemethoxycurcumin	↓ P-gp expression ↓ MDR1	[132,138]
Honokiol and magnolol (isolated from *Magnolia officinali*)	↓ MDR1 ↓ P-gp expression	[113,138]
Schisandrin A (Deoxyschizandrin)	↓ P-gp expression ↓ MDR1 ↓ PKC	[104,105,113,133,137,138,146]
Schisandrin B (Sch B)	↓ P-gp expression and P-gp mediated efflux of Dox. ↓ MRP1	[104]
Triptolide	↓ MDR1 ↓ MRP1 protein expression	[138]
Pyranocoumarins	↓ P-gp expression ↓ MDR1 mRNA expression	[133,138]
Ginger phytochemicals (6-Gingerol,10- Gingerol)	↓ P-gp expression ↓ MRP1	[114,138]
Ginger phytochemicals (6-gingerol, 10-gingerol, 4-shogaol, 6-shogaol, 10-shogaol, and 6-dehydrogingerdione)	↓ GSTπ ↓ MRP1	[221]
*Alisma orientalis*	↓ P-gp expression	[252]
*Piper methysticum*	↓ P-gp expression	[252]
Guggulsterone	↓ P-gp expression ↓ MRPs	[113,114,134,252]
Phenolic diterpenes	↓ P-gp expression	[252]
Vincristine	↓ P-gp expression	[252]
5-Bromotetrandrine	↓ P-gp expression	[133]
Abietane diterpene	↓ P-gp expression	[133]
Amooranin	↓ P-gp expression	[133]
Baicalein and derivatives	↓ P-gp expression ↓ MRPs	[103,132,133,134,143,146]
Bitter melon extract	↓ P-gp expression	[133]
Bufalin	↓ P-gp expression	[133]
Cannabinoids	↓ P-gp expression ↓ BCRP ↓ MRPs	[133]
β-Carotene	↓ P-gp expression	[116,133]
Fucoxanthin	↓ GST	[201]
Catechins	↓ P-gp expression	[111,133,143]
Cepharanthine	↓ P-gp expression ↓ MRP1	[133]
Coumarins	↓ P-gp expression	[133]
Cycloartanes	↓ P-gp expression	[133]
Didehydrostemofolines	↓ P-gp expression	[133]
Eudesmin	↓ P-gp expression	[133]
Euphocharacins A-L	↓ P-gp expression	[133]
Ginkgo biloba extract	↓ P-gp expression ↓ MRP1	[133]
Grapefruit juice extracts	↓ P-gp expression	[133]
Hapalosin	↓ P-gp expression	[133]
Hypericin and hyperforin	↓ P-gp expression ↓ BCRP	[133,262]
Isoquinoline alkaloid, isotetrandrine	↓ P-gp expression	[133]
Isostemofoline	↓ P-gp expression	[133]
Jatrophanes	↓ P-gp expression	[133]
*Kaempferia parviflora* extracts	↓ P-gp expression ↓ MRP1	[133]
Kavalactones	↓ P-gp expression	[133]
Ningalin B and derivatives	↓ P-gp expression	[133]
Opiates	↓ P-gp expression	[133]
Piperine	↓ P-gp expression ↓ BCRP ↓ MRPs ↓ ABC transporter genes (ABCB1, ABCG2, and ABCC1)	[133,134,135,136]
Polyoxypregnanes	↓ P-gp expression	[133]
Sesquiterpenes	↓ P-gp expression	[133,146]
Tenulin	↓ P-gp expression	[107]
Sinensetin	↓ P-gp expression	[133,146]
Taxane derivatives	↓ P-gp expression	[133]
Terpenoids	↓ P-gp expression ↓ BCRP	[133] [262]
Tetrandine	↓ P-gp expression	[133]
Vitamin E TPGS	↓ P-gp expression	[133]
3′-4′-7-Trimethoxyflavone	↓ BCRP	[133,143]
6-Prenylchrysin	↓ BCRP	[133,143]
Eupatin	↓ BCRP	[133]
Daizein	↓ BCRP	[133]
Hesperetin	↓ BCRP	[133,143,145]
Plumbagin	↓ BCRP	[133]
Resveratrol	↓ BCRP ↓ P-gp expression ↓ HIF-1α ↓ GST mRNA expression	[114,133,140,253]
Rotenoids	↓ BCRP	[133]
Stilbenoids	↓ BCRP	[133]
Tectochrysin	↓ BCRP	[133,143]
Tetrahydrocurcumin	↓ BCRP	[133]
Ligustrazine	↓ Expression of P-gp	[113]
Sophocarpidine	↓ Expression of P-gp	[113]
Strychnine	↓ Gene and protein expression of MRP	[113]
Three hydroxyl soy isoflavone	↓ MRP, MDR1, MRP2	[113]
Ecteinascidin	↓ P-gp expression	[113]
Ecteinascidin 743	↓ P-gp expression ↑ Cellular accumulation of DOX/VCR in P-gp-overexpressed cervix cells	[132]
7,3′,4′-trihydroxyisoflavone	↓ mRNA expression of MRP, MDR1, and MRP2	[113]
Paeonol (extracted from the dry velamen of *peony* or any part of *Cynanchum paniculatum*)	↓ P-g expression ↓ MDR1 ↓ MRP ↓ LRP	[113]
Oroxylin A-7-glucuronide	↓ MDR1 ↓ P-g expression	[113]
3′,4′,5′,5,7-pentamethoxyflavone (PMF) and derivatives of *epimedium*	↓ MDR1 ↓ P-g expression	[113]
Osthole (isolated from *Fructus Cnidii*)	↓ P-g expression	[113]
Praeruptorin A (extracted from *Radix Peucedani*)	↓MDR1 and P-gp mRNA	[113,146]
Diphyllin	↓ P-gp expression	[113]
Emodin	↓ P-gp expression ↓ MRP1 ↓ GSTπ ↓ Topo IIβ ↓ HIF-1α	[113,144,215,216,254,255,256]
Psoralen	↓ P-g expression	[113,144]
Gypenoside	↓ BCRP ↓ P-gp expression ↓ MRP1	[113]
Allicin	↓ MDR1 ↓ P-g expression	[113]
Taccalonolide A and B (extracted from *Tacca chantrieri*)	↓ P-g expression	[113]
Oridonin	↓ P-gp expression ↓ GSTπ ↓ LRP1	[113,144,222]
Ursolic acid (found in *Rosmarinus officinalis*)	↓ P-gp expression	[113,114]
Sipholenol A (found in *sponge Callyspongia siphonella*)	↓ P-g expression	[113,132,146]
Cantharidin (extracted from *Mylabris phalerata Pallas* or *Mylabris cichorii* L.)	↓ P-g expression	[113]
Beta-Elemene (isolated from *Aeruginous Turmeric rhizome*)	↓ P-g expression ↓ MRP	[113] [144]
As2O3, or white arsenic Arsenic Trioxide	↓ P-gp expression ↓ MRP	[113,144]
Artemisinin	↓ P-gp expression	[144]
Artesunate	↓ P-gp expression	[144]
Baicalin	↓ P-gp expression ↓ MRP1	[144]
Berberine (isolated from ancient Chinese herb *Coptis chinensis* French)	↓ P-gp expression ↓ABCG2	[144] [262]
Carnosic acid (Rosemary)	↓ P-gp expression	[114,134,144]
Chelerythrine	↓ P-gp expression	[144]
Gambogic acid	↓ P-gp expression	[144]
Neferine	↓ P-gp expression	[144]
Oxymatrine	↓ P-gp expression	[144]
Peimine	↓ LRP	[144]
Sodium norcantharidate	↓ P-gp expression ↓ MRP	[144]
*Brucea Javanica*	↓ P-gp expression ↓ MRP	[144]
Cinobufacini	↓ P-gp expression ↓ MRP1	[144]
Grape seed polyphenols	↓ P-gp expression	[103,144]
Hyaluronate Oligomers	↓ P-gp expression ↓ MRP	[144]
Jew ear	↓ P-gp expression ↓ MRP	[144]
*Radix notoginseng*	↓ P-gp expression	[144]
*Rhizoma pinelliae*	↓ P-gp expression	[144]
Realgar	↓ P-gp expression	[144]
*Thallus laminariae*	↓ P-gp expression	[144]
Algerian propolis	↓ transport function of P-gp-pump	[268]
Dihydroptychantol A (isolated from *A. angusta*)	↓ P-g expression	[252] [269]
Riccardin F (isolated from *P. intermedium*)	↓ P-gp expression	[269]
Riccardin D	↓ Topo II ↓ P-gp expression	[237]
Andrographolid	↓ P-gp expression	[105]
Parthenolide	↓ Pgp expression	[105]
Rhei Rhizoma, Scutellariae Radix, Poria, Zizyphi Fructus, Zingiberis Rhizoma, Asiasari Radix, Sophorae Radix (herbal extract)	↓ P-gp expression	[105]
Tripterygium wilfordii	↓ P-gp expression ↓ EGFR	[105]
Shenghe Powder (consisting of Radix codonopsis pilosulae, Radix pseudostellariae, Radix scrophulariae, Rhizoma atractylodis macrocephalae, and 6 additional herbs)	↓ P-gp expression	[105]
Shen-qi-jian-wei Tang	↓ MDR1 ↓ LRP	[105]
Yu Ping Feng San (YPFS) (Astragali Radix, Atractylodis Macrocephalea Rhizoma, and Saposhnikoviae Radix)	↓ ATP-binding cassette transporters ↓ GST	[219]
Chinese supplement energy and nourish lung (SENL) herbs (ginsenoside Rg1, ginsenoside Rb1, ginsenoside Rg3, astragaloside IV, ophiopogonin D, and tetrandrine)	↓ GSTπ	[220]
Icaritin	↓ P-gp expression	[132,134]
Icariin	↓ P-gp expression	[132]
Sesquiterpene ester 1	↓ P-gp expression	[132]
Celafolin A-1	↓ P-gp expression	[132]
Celorbicol ester	↓ P-gp expression	[132]
Demethoxycurcumin	↓ P-gp expression	[132]
Euphomelliferine	↓ P-gp expression	[132]
Euphodendroidin D	↓ P-gp expression	[132,146]
Pepluanin A	↓ P-gp expression	[132,146]
Sipholenone E	↓ P-gp expression	[132,146]
Siphonellinol D	↓ P-gp expression	[132]
GUT-70 (From *C. Brasiliense*)	↓ P-gp expression	[132]
Lamellarin I	↓ P-gp expression	[132]
Wogonin	↓ P-gp expression ↓ MRP1	[132]
Aposterol A	↓ P-gp expression ↓ MRP1	[132]
Fumitremorgin C	↓ BCRP ↓ P-gp expression ↓ MRP1	[109,132,162]
Tryprostatin A	↓ BCRP	[132]
Terrein	↓ BCRP	[132]
Lamellarin O	↓ BCRP ↓ P-gp expression	[132]
Secalonic acid D	↓ BCRP ↓ P-gp expression ↓ MRP1	[132]
Quinine and its isomer quinidine	↓ P-gp expression	[134]
Reserpine and yohimbine (isolated from *Rauwolfia serpentine*)	↓ BCRP ↓ P-gp expression	[134]
Bromocriptine ergot alkaloid	↓ P-gp expression	[134]
β-Sitosterol-O-glucoside	↓ P-gp expression	[134]
cardiotonic steroid 3	↓ P-gp expression	[134]
Menthol	↓ P-gp expression	[134]
Aromadendrene	↓ P-gp expression	[134]
Citronellal	↓ P-gp expression	[134]
Citronellol	↓ P-gp expression	[134]
Carnosol	↓ P-gp expression	[114,134]
Limonin	↓ P-gp expression	[134]
Kaempferide	↓ BCRP ↓ P-gp expression	[143,145]
Diosmin	↓ P-gp expression	[145]
Daidzein	↓ BCRP	[143,145]
Tanshinone microemulsion	↓ P-gp expression	[103]
Tea polyphenol	↓ P-gp expression	[103]
Stemocurtisine	↓ P-gp expression	[134]
Stemofoline	↓ P-gp expression	[134]
Oxystemokerrine	↓ P-gp expression	[134]
Amurensin G (from Vitis amurensis)	↓ P-gp expression	[270]
Sakuranetin	↓ P-gp expression	[143]
Floretin	↓ P-gp expression	[143]
Fisetin	↓ P-gp expression ↓ GST	[143,218]
Xanthohumol (derived from *Humulus lupulus*)	↓ mRNA expression of P-gp, MRP1, MRP2 and MRP3	[143]
Silybin (isolated from *Silybum marianum*)	↓ MRP1 ↓ P-gp expression	[143]
Sophoraisoflavone A	↓ MRP1	[143]
LANGDU (a traditional herbal medicine)	↓ P-g expression	[262]
Tanshinone IIA (isolated from *Salvia miltiorrhiza*)	↓ MRP1 ↓ BCRP ↓ P-g expression	[262]
Auraptene (grapefruit)	↓ P-g expression	[114]
Nimbolide	↓ P-gp gene	[164]
*Marsdenia tenacissima*	↓ P-g expression ↓ ABCG2 ↓ MRP1	[106]
Taxifolin	↓ ABCB1 ↓ P-gp expression	[111]
*Heterotheca inuloides* Cass.	↓ MDR1 ↓ MRP1 ↓ BCRP	[112]
Saikosaponin D	↓ MDR1 gene ↓ P-gp expression	[271]
Kanglaite (isolated from *Coix lacryma-jobi*)	↓ MDR1 ↓ MRP2 ↓ BCRP	[272]
*Astragalus membranaceus* polysaccharides Astragaloside II, another component from *A. membranaceus*	↓ P-gp expression ↓ MDR1	[273]
Wilforine	↓ P-gp expression	[110]
*Boswellia serrata* extracts 3- O-acetyl-11-keto-β-boswellic acid (AKBA), the major active ingredient of the gum resin from Boswellia serrata and Boswellia carteri Birdw	↓ P-gp expression	[274]
Pervilleine F	↓ P-gp expression	[146]
Ellipticine	↓ P-gp expression	[146]
Cnidiadin	↓ P-gp expression	[146]
Conferone	↓ P-gp expression	[146]
Rivulobirin A	↓ P-gp expression	[146]
Dicamphanoyl khellactone (*DCK*)	↓ P-gp expression	[146]
Cannabidiol	↓ P-gp expression	[146]
Taccalonolides A	↓ P-gp expression	[146]
Jolkinol B	↓ P-gp expression	[146]
Portlanquinol	↓ P-gp expression	[146]
Dihydro-β-agarofuran	↓ P-gp expression	[146]
Pentadeca-(8,13)-dien-11-yn-2-one	↓ P-gp expression	[146]
Silibinin	↓ P-gp expression	[146]
Nirtetralin	↓ P-gp expression	[146]
Cordycepin	↓ P-gp expression	[107]
Nuciferine	↓ HIF-1α	[275]

## 4. Synthetic Compounds in Reversing Chemo-Resistance

Natural products are the main source of developing new drugs, and highly active phytochemicals are suitable for further modifications to formulate more effective analogs and prodrugs [260]. As mentioned, curcumin is one of the herbal remedies that has been proved to have a broad spectrum of therapeutically significant properties [253]. Interestingly, four pyrimidine-substituted curcumin analogs were tested as promising P-gp inhibitors. One of these synthetic compounds exhibited high potency and low toxicity in reversing the P-glycoprotein- mediated MDR in paclitaxel-resistant human breast cancer cells [276]. As a solution for the poor water solubility of curcumin, novel pyrazolo derivatives have been designed. They remarkably increased the sensitization of multidrug-resistant cells to doxorubicin and revealed a chance of developing potent P-gp antagonists [251]. Moreover, shikonin derivatives (acetyleshikonin and acetoxisovaleryshikonin) have been reported to trigger uptake and reduce efflux of anticancer agents in malignant carcinoma cells [277]. Additionally, novel isomers of methylated epigallocatechin and catechin have shown specific inhibition for P-gp and reversed the MDR in cancer cells [278]. Parthenolide and 5-fluorouracil conjugates have been synthesized and tested on drug-resistant hepatocellular carcinoma cells. The most active compound exhibited a high ability to inhibit MDR1, ABCC1, and ABCG2; as a result, it increased drug accumulation and induced apoptosis [275].

## 5. Targeting Non-Apoptotic Cell Death Using Natural Products

### 5.1. Targeting Necroptosis

The dynamic balance among cell proliferation, differentiation and death is of great significance in maintaining tissue homeostasis [263,264]. For a long time, apoptosis has been considered to be the single pathway in programmed cell death (PCD) [238,261]. In contrast, necrosis refers to the process that cells swell, rupture, and then release cellular contents and proinflammatory molecules in response to the overwhelming stress [262,265]. Necrosis is regarded as an uncontrollable process, and therefore it is highly challenging to identify small molecules that can interfere with this process [262,265]. In the late 1980s, cells under the treatment of tumor necrosis factor (TNF) were found to be characterized with either apoptotic or necrotic features in a cell type-dependent manner [266]. In 2004, Thompson et al. [252] discovered that alkylating agent N-methyl-N^0^-nitro-N-nitrosoguanidine initiated cell necrosis depending on the expression of poly ADP-ribose polymerase (PARP), suggesting that necrosis could be regulated by small molecules. After one year, the concept “necroptosis”, a combination of “necrosis” and “apoptosis”, was proposed [267]. Since apoptosis-resistant cases were reported in clinical cancer therapy [256], it was important and urgent to search for a new pathway to induce the death of cancer cells, while necroptosis is a promising alternative [268,269]. Therefore, intensive efforts and attention have been paid to investigate the mechanism of necroptosis and clarify its relationship to cancer therapy [270,271,272]. Recent studies indicated that necroptosis could be initiated by activation of specific receptors on cell membrane and regulated by cytokines and small molecules [273,274,279,280,281,282]. While the relationship between necroptosis and cancer has not been clarified explicitly, both efforts are being focused on the factors relevant to the induction and inhibition of necroptosis [283,284,285]. Thus, a better understanding of the mechanism of necroptosis and its modulators is important to develop novel strategies for cancer therapy.

Cell exposure to stress elicits the inherent response of cell death, which involves cell removal from the microenvironment. Necroptosis is among the several different molecular pathways and mechanisms that usually control cell death.

Recently, a necrotic process has been discovered, which is characterized by changes in the plasma membrane leading to inflammatory cell function, and it has recently been discovered that it is controlled by genes. Cytokines, pathogens, ischemia, heat, and radiation trigger various pathways (such as death receptors, kinase cascades, and mitochondria) that can induce necroptosis [286,287,288].

Over the last 20 years, it was found that nonapoptotic cell death can occur in a regulated fashion [289]. Chemical biology approaches were central to establishing this new paradigm, as exemplified by the study of necroptosis [290]. Necroptosis is classified as a programmed cell death in the absence of morphological traits of apoptosis or autophagy [286,287,288,291,292,293,294]. Emerging evidence shows that necroptosis can be disturbed in many human cancers [295,296,297,298,299,300,301,302,303,304,305,306,307,308,309,310].

Necroptosis describes regulated cell death, typically caused as a result of inflammatory pathway activation. It is a programmed means of cellular demise; the etiology is a physical insult that stimulates a signalling pathway autonomous of caspase [311]. Tumor necrosis factor a, receptor interacting protein kinase 3, and caspase-8 have been the subjects of considerable research in order to identify the mechanisms underlying necroptosis at a molecular level. The process may be instigated by the TNF superclass, Toll-like and interferon receptors, respectively. Necroptosis involving the former is well-delineated [287,295,312,313,314,315].

Previous studies confirmed that necroptosis was an invitation to inflammation [316]. Damage-associated molecule patterns (DAMP) are capable of recruiting primary immune cells to the necroptotic cells, followed by phagocytosis and termination of cell death signaling [317]. Furthermore, inflammation is inclined to tumorgenesis [318] and promotes cancer progression, metastasis, and increasing drug resistance [319].

Necrotic cells are eliminated from the immune system through pinocytosis or cell drinking, which is mediated by macrophages, a subcellular component of macrophages [320]. The propagation process of necroptosis is irreversible; therefore, more attention should be paid to the discovery of inhibitors to block the occurrence and execution of necrosis.

Metastasis is the primary cause of morbidity and mortality in cancer patients [321]. It was reported that shikonin dramatically reduced the metastasis of osteosarcoma C6 and U87 glioma cells due to its induction of necroptosis [310,322]. Moreover, ROS play a significant role in the migration and invasion of cells through regulation of cytoskeleton dynamics and adhesion molecules [310,323]. The generation of ROS promotes the execution of necroptosis. In other studies, RIPK3 was supposed to modulate the ROS levels in the case of necroptosis [323]. Thus, necroptosis has a close relationship with the metastasis of cancer cells.

Necroptosis serves as a back-up for apoptosis. Before the concept of necroptosis was suggested, a prevailing trend in cancer chemotherapy was to discover molecules that can induce apoptosis [324]. However, apoptosis inducers cannot work in some circumstances, such as drug resistance due to the activation of alternative compensatory pathways, upregulation of drug transporters, and multidrug resistance [324]. For example, proapoptotic agents can induce necroptosis in cancer cells when apoptosis is blocked [325]. When a diphtheria-based fusion toxin applied with demethylating agents, it can synergistically trigger cancer cell death and overcome apoptosis resistance by inducing necroptosis [324]. Therefore, necroptosis can serve as an alternative method in cancer therapy in place of various pathways of apoptosis.

Necroptosis may contribute to cancer progression. Necroptotic cells are characterized by cell membrane leakage and the release of molecular patterns related to intracellular damage, which result in inflammatory responses and related side effects [317]. Subsequently, they may promote tumor progression, promote tumor cell proliferation and survival, as well as tumor angiogenesis, invasion, and metastasis, and adversely affect tissues, which is not conducive to treatment [326]. Conversely, damaged and aging tissues may also promote the metastasis of cancer cells [326].

Necroptosis is not universally sensitive in tumor cell lines. Due to different environments (such as the availability of oxygen or nutrients), different cancer cell lines have different sensitivities to necroptosis [327]. Therefore, chemotherapy based on necroptosis is only effective for a limited number of tumor cell lines [327,328].

In summary, necroptosis of cancer cells could be initiated by various stimuli and through different pathways. Necroptosis is associated with tumor metastasis, which is the major cause of morbidity and mortality in cancer patients in clinic. Therefore, induction of necroptosis is an effective strategy in clinical cancer therapy. Necroptotic cells could potentially induce intrinsic and adaptive immune response and thus mediate efficient antitumor immunity. In addition, necroptosis can serve as a back-up for apoptosis-resistant circumstances, which is capable of overcoming the obstacle of drug resistance that is common but intricate in clinical cancer therapy. For those apoptosis-resistant cases, necroptosis inducers could sensitize tumor cells to death, which provides another way to overcome drug resistance when common therapy fails. Thus, drug development of necroptosis inducers deserves more attention. However, the release of cytokines also induces inflammation which would be harmful to the tissue and in turn tissue damage could facilitate the metastasis of tumor cells. However, necroptosis is accompanied by the release of cytokines and induces inflammation, which is harmful to the tissue and in turn tissue damage can facilitate the metastasis of tumor cells. Additionally, it should be noted that necroptosis is not widely sensitive in cancer cells in that apoptosis is the main cause of cancer death. Therefore, necroptosis inducers are mostly favorable in apoptosis-resistant cases in clinical cancer therapy. Currently, several antitumor agents have been verified to act as necroptosis inducers, while more medicinal chemistry efforts are required to discover new inducers with drug-like properties.

Thus far, a series of necroptosis inducers have been identified, paving the way for studying new patterns of cancer death and providing new treatment tools. However, most studies are conducted in vitro, and the in vivo efficacy of necroptosis inducers and their tumor-killing selectivity still need to be further explored. In addition, there is still a lack of necroptotic markers in the body. Therefore, there is a great need to discover new necroptotic markers and study their effects on cancer cell selectivity. With a better understanding of the mechanism of cancer cell necroptosis, targeting necroptosis will become an effective strategy for cancer treatment.

As a genetically regulated mode of cellular demise, necroptosis is an essential mechanism contributing to human pathologies. A vital method of eradicating cells, the process is linked with malignancy advancement and dissemination, and immunosurveillance. Targeting the processes underlying necroptosis with small molecular modulators appears to be an encouraging strategy in oncology therapeutics; benefits include the circumvention of apoptosis resistance and ongoing anticancer immunity integrity. Thus, in order to design de novo approaches for tumor treatment, an improved comprehension of necroptosis and its governing molecules is required.

While numerous reviews focused on the apoptotic pathways, extrinsic and intrinsic alike, few articles concentrated on necroptosis induced by natural compounds in tumor cells [329,330,331,332,333,334]. Table 2 shows a list of natural compounds with an effect on non-apoptosis cell death in tumor cells.

Natural compounds originating from plants, microorganisms, and marine life forms were widely shown to display anti-carcinogenic, anti-proliferative, and anti-survival effects inducing tumor cell death via various pathways, including necroptosis, leading to cell death [329,330,331,332,333,334,335,336,337,338,339,340,341,342,343].

An alkaloid extracted from traditional Chinese herb Sophora flavescens, called matrine, was reported to induce necroptosis in cholangiocarcinoma cells, which was different from its classic apoptosis-inducing effects in other cancer cell lines [344]. Mz-ChA-1 and QBC939 cells (cholangiocarcinoma cells) treated with matrine were characterized by extensive organelles, plasma membrane rupture, and integral nuclei which coincided with the morphology of necroptotic cells [344]. Additionally, cells that died due to the treatment of matrine could not be rescued by the addition of pan-caspase inhibitor 17, but they could be rescued by necroptosis inhibitor 2 [344]. Further studies confirmed that matrine induces necroptosis through the formation of RIPK1/RIPK3/MLKL complex [344]. Osmotic pressure and release of ROS also facilitated the necroptotic process [344]. More interestingly, matrine was able to upregulate the expression of RIPK3 in Mz-ChA-1 cells, making it possible to treat cholangiocarcinoma with a low expression of RIPK3 [344]. This work potentially provided a new strategy to deal with the apoptosis resistance in cholangiocarcinoma therapy [344].

Neoalbaconol, extracted from Albatrellus confluens, was reported to trigger several kinds of cell death [345]. NA downregulates the E3 ubiquitin ligase, resulting in a decrease in RIPK1 ubiquitination. Therefore, an increase in the expression level of RIPK1 was observed, which also activated the transcription of TNF-a [345]. In addition, NA causes RIPK3-mediated ROS generation, which contributes to cell death [345]. Therefore, it can be concluded that NA can induce necroptosis by activating TNF-α and RIPK3-dependent ROS production [345].

Shikonin, extracted from the traditional Chinese herb “Zicao”, was originally used to treat wound healing because of its anti-inflammatory and antimicrobial properties [346]. It was reported to kill tumor cells by inducing apoptosis [346]. Huang et al. [322] found that losses in plasma membrane integrity and intact nuclear membranes in glioma cells were observed in cells treated with shikonin directly by electronic transmission microscopy. Shikonin-induced C6 and u87 glioma cells can be rescued by necroptosis inhibitor 2, but are unaffected by treatment with the caspase inhibitor 17. Additionally, an increased expression of RIPK1 was observed after treatment with shikonin [322]. All these facts indicate that shikonin induces necroptosis through RIPK1 activation.

Emodin, an anthraquinone derivative extracted from traditional Chinese medicine Rheum palmatum, has been used to treat various diseases due to its antitumor, anti-inflammation, anti-metastasis, and immunosuppressive effects, but its mechanism of action remains unclear [347]. Zhou et al. [348] found that cells treated with emodin showed increased levels of RIPK1, RIPK3, and TNF- a. In addition, the combined use of emodin and necroptosis inhibitor 2 or 15 reduced the release of lactate dehydrogenase (LDH) [348]. Hematoxylin-eosin (H&E) staining of tumor tissue isolated from mice treated with emodin showed an obvious necrotic effect [348]. Further studies indicated that TNF-a, RIPK1, RIPK3, and MLKL in tumor tissues were upregulated when treated with emodin, indicating that it could inhibit glioma growth in vivo through necroptosis [348].

Ungeremine is an alkaloid extracted from Ungernia minor, which shows effective cytotoxicity in drug-resistant cancer cell lines and upregulates RIPK3 levels [349]. Co-treatment with Necroptosis Inhibitor 2 reduced the cytotoxicity of ungeremine, and flow cytometry analysis showed that 13.1% of the cells were necrotic [349].

Several natural compounds were shown to induce necroptosis in human tumor cells. For example, shikonin, a naphthoquinone derived from Lithospermum erythrorhizon Siebold & Zucc. (Zicao), was found to be the first natural product capable to induce both apoptosis and necroptosis in human several cancer cell lines [310,322,350,351,352,353,354,355,356,357,358,359,360,361]. Shikonin-treated cells showed morphological alterations distinct from those occurring in apoptosis or autophagic cell death [310,322,350,351,352,353,354,355,356,357,358,359,360,361]. The loss of plasma membrane integrity was one of the morphologic characteristics of necrotic cell death [310,322,350,351,352,353,354,355,356,357,358,359,360,361]. Shikonin exerts a dramatic anticancer effect on both primary and metastatic osteosarcoma by inducing RIPK-1 and 3- dependent necroptosis [310,351]. The size of primary osteosarcoma tumors and lung metastases was significantly reduced by shikonin treatment accompanied the elevated levels for RIPK-1 and RIPK-3 proteins [310,351]. Shikonin was observed to induce necroptotic cell death in tumor cells involving drug- and apoptosis-resistant cancer cells that overexpress multidrug resistance protein-1, P-glycoprotein, and other proteins [310].

Shikonin exerts effective cytotoxicity on human multiple myeloma borate-resistant KMS11/BTZ cells [353]. Shikonin can circumvent the drug resistance displayed by tumor cells and mediated by P-glycoprotein, BCL-2, and BCL-xL by inducing necroptosis [355,356,357,358,359]. Naturally occurring shikonin analogues (deoxyshikonin, acetylshikonin, isobutyrylshikonin, β, β-dimethylacryloylshikonin, isovalerylshikonin, α-methyl-n-butylshikonin) induce necroptosis in resistant tumor cells overexpressing MDR1 and BCRP1 [322,352,353,354,355,356,357,358].

Staurosporine, a protein kinases inhibitor, induces necroptosis in leukemia cells when caspase activation is inhibited [362]. It was recently determined that the TNF-related apoptosis-inducing ligand (TRAIL) induces necroptosis via RIPK-1-/RIPK-3-dependent PARP1 activation, consequently turning PARP1 activation into an effector mechanism downstream of RIPK-1 [363,364]. Obatoclax (GX15-070), a small-molecule inhibitor of antiapoptotic BCL-2 proteins, was shown to induce autophagy-dependent necroptosis in glucocorticoid-resistant acute lymphoblastic leukemia in childhood [365,366,367]. It was further observed to induce an assembly of necrosomes on autophagosomal membranes [365]. It increased the physical association of ATG5 with RIPK-1 and 3 [365]. Furthermore, photodynamic therapy by incorporating 5-aminolevulinic acid was noted to trigger RIPK-3-mediated necrosis in glioblastoma cells [368].

Piperlongumine (a natural constituent of the fruit of Piper longum), and taurolidine (a derivative of taurine) were observed to inhibit in vitro and in vivo cancer cell growth [369]. Piperlongumine can selectively kill specific types of cancer cells over normal cells [369]. It inhibits the growth of xenografted human malignant breast tumors in vivo [369]. Taurolidine induces cytotoxicity in different cancer cells in vivo and in vitro [369]. The effect of piperlongumine and taurolidine is dependent on the stimulation of programmed cell death by autophagy via a redox-directed mechanism, in addition to apoptosis and necroptosis [369]. A decrease in the ROS levels markedly reduces the AA005-mediated cell death in SW620 cells [370]. Additionally, the RIPK-1 repression by NEC-1 prevents the translocation of the apoptosis-inducing factor and partially destroys AA005-induced cell death in SW620 cells, suggesting a role for the necroptotic pathway [370].

Eupomatenoid-5 (Eup-5) was found to induce cell death in renal cancer 786-0 cells in addition to human breast tumor MCF-7 cells [371]. In MCF-7 cells, Eup-5 caused phosphatidylserine externalization and caspase activation, while 786-0 cells treated with Eup-5 showed characteristics of the programmed necroptosis process [371]. Green tea polyphenols were shown to induce BAX and BAK activation, cytochrome c release, caspase activation, and necroptosis of human hepatocellular carcinoma Hep3B cells [370]. Interestingly, in Hep3B cells lacking BAX (BAX−/−) or BAK (BAK−/−) expression, cytochrome c release and necroptosis were reduced [372].

Concurrent necrosis and apoptosis have been demonstrated following parthenolide in human acute promyelocytic HL-60 and Jurkat T lymphoma cells, respectively [373]. Membrane breach and swift necrotic cell demise resulted from the engagement of parthenolide with the cell membrane [374]. In human MDA-MB231 breast cancer cell lines, necrosis induced by parthenolide was diminished by NAC and NEC-1, thus inferring that ROS and RIP-1 stimulation may be relevant [375].

### 5.2. Targeting Autophagy

Autophagy, formerly termed macroautophagy, is a further essential mode of malignant cell demise following chemotherapy. It is characterized by self-destructing intracellular proteins and organelles within the lysosome. The process is flexible and necessary for cellular equilibrium, stimulating cellular demise without the presence of apoptotic moderators but when vital autophagy-governed genes, e.g., *ATG5* [376] and *BECN1* [377], are evident.

Autophagy is the process by which cells forms double-membrane autophagic vesicles (AV), which isolate organelles and proteins and target them for degradation in the lysosome. Although it was originally viewed as a “bulk degradation” process activated by cellular starvation, new findings demonstrate that autophagy can also be a highly selective quality-control mechanism that regulates levels of specific organelles and proteins [378,379,380,381].

Exhaustion of nutrients is maintained by the highly conserved adaptive process called autophagy. This process is exploited by solid tumor cells to achieve homeostasis through cytoplasmic waste metabolism to satisfy growing demand for nutrients. This promotes development of cellular stressors such as inflammation, hypoxia, and tumor progression. Different solid tumors have been found to have poor prognosis depending on the extent of autophagy [382,383,384,385].

Autophagy can be categorized as microautophagy, macroautophagy, and chaperone-mediated autophagy; all classes frequently enhance the proteolytic eradication of cytosolic constituents within the lysosome [386,387,388]. Contemporary work has concentrated particularly on macroautophagy, including delineation of its properties at cellular and molecular levels, and its potential use in antitumor treatment [389,390].

In cancer, autophagy may play a role in limiting the earliest stages of tumorigenesis; however, there is growing evidence that, in established cancers, autophagy can help cope with intracellular and environmental stresses, such as hypoxia, nutrient shortage, or cancer therapy, thereby favoring tumor progression as a survival mechanism. In this context, it is becoming increasingly clear that autophagy inhibition could improve therapeutic outcomes for patients with advanced cancer.

In contrast to normal cells, tumor cells frequently exhibit impaired autophagic processes, although the relevance of autophagy in malignancy is contradictory to that in the normal cellular state. Current evidence implies that autophagy may both advance and hinder tumor development by enhancement of neoplastic cell longevity and a cancer-suppressive action in non-malignant cells, respectively [391]. Elimination of structurally compromised proteins and organelles may inhibit oncogenesis and enable existing cancers to survive nutrient or oxygen lack during their growth [392].

Our current understanding is that the autophagy pathway consists of at least seven steps [393,394,395,396,397,398,399,400,401]. The conserved autophagy genes (ATG genes) regulate steps 1 to 5:Step 1: “The Unc-51-like kinase protein kinase complex”: regulates initiation of AV formation.Step 2: “The VPS34 lipid kinase complex”: prepares the membrane for curvature.Step 3: “LC3 family conjugation cascade”.Step 4: “Cargo loading through autophagy cargo adaptors”.Step 5: “AV maturation”.

The genes that are common to other endosomal/lysosomal pathways promote steps 6 and 7.
Step 6: “AV–lysosome fusion”.Step 7: “Lysosomal degradation and recycling of AV cargo”.

Although these seven steps of autophagy are well documented, it is expected that additional regulators will be discovered. New, novel regulators of autophagy were recognized using an siRNA screen in a pancreatic cancer cell line. Two of the promising candidates, MAGUK p55 subfamily member 7 (MPP7) and cytosolic malate dehydrogenase 1 (MDH1), were observed to have roles in forming the autophagosome [402]. Many of these steps in the autophagy pathway represent potentially druggable targets providing ways to both positively and negatively influence autophagy. As autophagy is a complex multistep mechanism, understanding the details of autophagy is critical to developing effective compounds and therapies to modulate autophagy potently and specifically.

Our current understanding is that the autophagy pathway consists of at least seven steps [403,404,405,406,407,408,409,410,411]. The conserved autophagy genes (ATG genes) regulate steps 1 to 5: Step 1: The Unc-51-like kinase protein kinase complex regulates the initiation of AV formation. Step 2: The VPS34 lipid kinase complex prepares the membrane for curvature. Steps 1 and 2 arrange intracellular membranes to form AVs via enriching the membrane for phosphatidylinositol 3 phosphate (PI3P). Step 3: LC3 family conjugation cascade. Step 4: Cargo loading through autophagy cargo adaptors. Step 5: AV maturation. The genes shared by other endosomal/lysosomal pathways facilitate steps 6 and 7: Step 6: AV–lysosome fusion. Step 7: Lysosomal degradation and recycling of AV cargo. Although these seven steps of autophagy are well documented, it is expected that additional regulators will be discovered. New, novel regulators of autophagy were recognized using an siRNA screen in a pancreatic cancer cell line. Two promising candidates, members of the MAGUK p55 subfamily 7 (MPP7) and cytosolic malate dehydrogenase 1 (MDH1), were observed to have roles in forming the autophagosome. MPP7 activates YAP1, induces autophagy, and MDH1 regulates ULK1 levels [412]. Many of these steps in the autophagy pathway represent potentially druggable targets providing ways to both positively and negatively influence autophagy. As autophagy is a complex multistep mechanism, understanding the details of autophagy is critical to developing effective compounds and therapies to modulate autophagy potently and specifically.

The value of autophagy in health and disease was recently emphasized when Yoshinori Ohsumi received the Nobel Prize in Medicine in 2016 for his work on advancing understanding of the genetic foundation of autophagy in yeast [413]. Multiple groups were involved in endeavors to demonstrate that [403].

A number of human pathologies are associated with autophagic anomalies, e.g., dementia, cardiovascular conditions, leishmaniasis, influenza, liver disorders and malignancy, e.g., hepatocellular carcinoma (HCC).

The fifth leading form of fatal hepatic malignancy worldwide, HCC, has a heterogeneous geographical prevalence, affecting males three-fold more than females. Current therapeutic interventions have a range of clinical outcomes; short survival times result from pharmaceutical resistance and adverse events.

Autophagy is one of the underlying disease processes that is targeted during the design of antitumor agents. Overall, overstimulated autophagy may precipitate a non-apoptotic type of programmed cell death (PCD), autophagic or type II PCD. Recent studies have inferred that malignant cell type II PCD can be initiated through autophagic interference, thus inhibiting cancer progression. Influencing associated signalling is therefore an encouraging route for the generation of de novo agents to combat drug-resistant tumor cells.

Natural polyphenolic substances, e.g., flavonoids and non-flavonoids, exert an antitumor action through the upregulation of cancer suppressors and autophagy mediated via signalling pathways that are both canonical (Beclin 1-dependent) and non-canonical (autonomous of Beclin-1). Angiogenesis and tumor dissemination in HCC have been shown to be targets of plant polyphenols; these influence numerous intracellular mediators and diminish HCC risk [404].

Some research, particularly from Kroemer’s group, has postulated that autophagy suppression would be detrimental to tumor therapy as it would diminish the responses of antitumor T cells [405,406,407]. In expiring malignant cells, autophagy is believed to be necessary for immunogenic cellular demise, giving rise to cellular identification by the immune system and stimulation of an efficacious immune reaction [408,409]. However, this research was conducted in extremely immunogenic experimental models [410], which may have influenced the data. In contrast, research employing murine tumor models of lower immunogenicity demonstrated comparable T cell responses between animals with and without effective autophagy; in the latter, genetic deletion of relevant genes or pharmacological autophagy inhibition with CQ was applied [411].

Progressing the concept of the necessity of autophagy for immunogenic cell eradication, Kroemer et al. surmised that autophagy, upregulated utilizing caloric restriction mimetics, could uplift anti-neoplastic immune responses [412], implying that autophagy should not be suppressed and that anticancer therapies should be potentially targeted towards its enhancement.

Thus, although here only contemplating anticancer immune reactions, inhibition of autophagy remains the subject of debate [414,415,416] and could have the opposite effect to that desired from first line treatment [417,418,419].

A further issue relating to targeting autophagy is that in patients, autophagy suppression would not be restricted to cancerous cells, thus causing toxicity from more widespread effects. For instance, research has demonstrated that in adult mice, global tissue knockout of *Atg7*, a critical autophagy gene, caused their ultimate demise as a consequence of grave neuronal dysfunction, loss of glucose metabolism, and enhanced vulnerability to infection [420].

Nevertheless, although extracting a vital constituent of the canonical pathway of autophagy from each cell may mimic the impact of a ‘flawless’ autophagy suppressor, this approach is at variance with clinical applications where a pharmaceutical agent would be less potent than total knockout of a key autophagy moderator.

Reinforcing this concept, long-term HCQ utilisation in rheumatological conditions and CQ therapy in certain malignancies as autophagy suppressors has not been associated with intolerable toxicity [421], showing that ongoing lysosomal autophagy inhibition is possible. Essentially, if malignant tissues have a higher reliance on autophagy than normal cells, an agent that gives rise to toxicity in the latter may have a clinically appropriate treatment period for efficacious tumor therapy. In fact, in the *Ag7* knockout model, KRAS-driven lung cancers were notably suppressed by gene deletion prior to evidence of neurotoxicity [420], confirming the potential presence of a therapeutic window for autophagy suppression in certain malignancies.

The precise function of autophagy in controlling malignant immune responses remains obscure and the controversy regarding autophagy suppression continues. Additional studies are merited to characterize its associated hazards and application whilst simultaneously heightening immune responses to obtain positive clinical endpoints. Markers denoting autophagy dependence may be useful for the recognition of individuals who would gain most benefit from treatment by autophagy inhibition.

Autophagy can help cancer cells to survive and withstand the action of therapeutic agents in the context of chemotherapy-activated cellular stress. A correlation between extensive autophagy and systemic therapy resistance, including androgen inhibition therapy in prostate cancer, has been highlighted by a number of clinical and preclinical models [422,423,424,425,426,427,428,429,430].

A very large body of literature supports the induction of autophagy during cancer therapy as a key resistance mechanism in multiple cancer types. Many of the stress-sensing pathways that induce autophagy are engaged by cancer therapies used in the clinic. Although complete and prolonged autophagy can produce cell death in vitro and in vivo, the major role of autophagy induction is to enable survival of the cancer cell during therapy. For example, cisplatin-induced autophagy was shown to play a role in tumor cell resistance to platinum-based therapy [431]. Cytotoxic chemotherapy, targeted therapy, and radiotherapy can all activate cytoprotective autophagy [432].

Emerging evidence suggests that autophagy is upregulated in tumor cells in response to various stresses, leading to tumor cells becoming resistant to chemotherapy. Therefore, targeting the autophagy signaling pathway may be an innovative strategy for human cancer prevention and combination therapy and for combating tumor-derived chemoresistance [433,434].

Increasing evidence suggests the possibility that autophagy may be a key actor in the evolution of drug resistance in tumor cells. Enhanced autophagy was demonstrated in individuals with melanoma in who neoplastic cells developed resistance to vemurafenib, a BRAF inhibitor, through an ER stress reaction [435]. Additionally, vemurafenib resistance induced by sustained melanoma cell line culture with the agent could be abolished via autophagy suppression [435]. Clinically, a patient with BRAF-mutant cerebral malignancy, in whom vemurafenib was at first successful but then hindered by drug resistance, had a positive outcome when prescribed CQ as an adjunctive therapy [421]. This case report demonstrated resensitisation of a malignancy by autophagy inhibition. Of note, was that the admixture of kinase and autophagy inhibitors, and not the latter alone, was beneficial for the ongoing suppression of cancer progression, suggesting that the clinical advantage pertains to surmounting drug resistance as opposed to engineering de novo responsiveness to sole treatment with an autophagy suppressor [421].

Additionally, the contribution of autophagy to multi-drug resistance, frequently to routinely used agents employed for challenging malignancies, has been inferred. In ovarian cancer, provocation of autophagy has underpinned resistance to paclitaxel, a cytotoxic agent [436]. Similarly, this process has been demonstrated to underlie cisplatin resistance in ovarian and oesophageal cancers, and occur as a result of low oxygen levels in lung tumors [437]. In primary chronic lymphocytic leukemia cells in patients, precipitation of autophagy owing to an ER stress reaction, as in melanoma, leads to resistance to cyclin-dependent kinase (CDK) inhibitors [438] and to resistance to HDAC inhibitors, e.g., tubastatin A, in cell lines derived from glioblastomas [439]. The connection between autophagy and chemotherapy resistance continues to grow and so autophagy, as a potential target for antitumor agents, will inevitably remain the subject of ongoing research [440,441,442,443,444].

Autophagy has additionally been associated with promotion of dormant cancer cell survival and may form a key enabler of cell growth recommencement [445]. Contemporary work on a malignant drosophila model demonstrated that quiescent neoplastic tissue from autophagy-deficient flies was initiated into growth when relocated into autophagy-competent organisms. These results implied that non-malignant cell independent autophagy in the environs’ adjacent cells is essential for growth reestablishment in quiescent neoplasia [446]. If comparable events take place in mammals, this study would indicate that endeavors to promote autophagy following a presumed positive outcome from tumor therapy may have the undesired consequence of precipitating recurrence by stimulating remaining dormant malignant tissue.

Autophagy may be an efficacious tumor escape mechanism; it is thought to contribute to resistance evolution in numerous malignancies, e.g., those originating in the central nervous system, melanoma, non-small cell lung cancer (NSCLC), and neoplasia of the bladder and thyroid. Concomitant autophagy inhibition therapy in these instances may diminish or rectify treatment resistance.

In the oncology sector, autophagy has conflicting and setting-reliant influences and so a ‘one size fits all’ strategy with therapies engineered to suppress or to promote this process in tumor treatment will not have a positive outcome. In view of this realization, the optimal approach could be to circumvent interference with autophagy altogether in this context. Nevertheless, changes to autophagy are inevitable, as it is impacted by numerous contemporary therapies. Furthermore, physiological triggers, particularly those that frequently exert diverse influences on malignant and normal cells, respectively, e.g., lack of nutrients or oxygen, will additionally impact the autophagic process in neoplastic tissue. Thus, it is important to delineate the consequences of these alterations and to attempt to customize treatments to suit specific contexts. To begin with, such therapies will potentially center on autophagy suppression; a major requirement is then to determine which patients would benefit from this strategy.

It has been discovered that a variety of compounds can induce tumor cell death by deregulating the signaling pathways that lead to autophagy [329,333,334,447,448,449,450,451,452,453] (Table 2). Melatonin promoted vincristine-resistant oral cancer cell autophagy and apoptosis and reduced the drug resistance of vincristine-resistant oral cancer cells by increasing the expression of microRNA [454].

An autophagy-mediated mode of cell death is generated by the application of a number of polyphenolic substances in a range of malignant cells [455]. A category of natural organic chemicals, polyphenols typically comprise numerous phenol structural moieties and are present in the diet. Several, e.g., rottlerin [456], curcumin [457], resveratrol [458], genistein [459,460], and quercetin [461] exhibit antitumor properties, influencing signalling pathways and promoting cell death through both apoptotic and autophagic processes [462,463]. These substances can stimulate type II PCD through a spectrum of mechanisms via canonical (Beclin-1-dependent) and non-canconical (autonomous of Beclin-1) autophagic avenues (212), and have the potential to be utilized as adjuncts to routine antitumor treatments.

Polyphenols additionally moderate autophagy in order to surmount or to reverse cellular multi-drug resistance. Apigenin notably enhances the responsiveness of doxorubicin-resistant BEL-7402/ADM cells, triggers miR-520b expression, and inhibits Atg7 [464]. Interference with autophagy gives rise to PARP-1 upregulation in ovarian tumor cells undergoing therapy with cisplatin; this is essential for cellular longevity. PARP-1 suppression, at mRNA and protein junctures, together with autophagy inhibition, can be achieved using luteolin, thus returning sensitivity to cisplatin [465]. An active flavone, scutellarin has been demonstrated to diminish cell cycle-associated Cdc2 and cyclin B1 protein expression, to activate apoptosis in PC3 cells, and to preserve resistance to cisplatin [465].

Naturally occurring polyphenols, flavonolignans, comprise flavonoid and lignin portions. Originating from silymarin extract, the flavonolignan, silibinin, displayed antigenotoxic, membrane-stabilizing, and anti-oxidant properties and triggered liver cell regrowth, inhibited fibrogenesis, and reduced the intrahepatic inflammatory response. The compound has been suggested to activate autophagy in HeLa cervical malignancy [466] and in MCF-7 breast tumor cell lines via the generation of LC3-II, Atg12-Atg5 and Beclin-1 upregulation [467]. Autophagy was promoted by silibinin in human fibrosarcoma HT1080 cells via ROS/p38/JNK pathway-induced p53 stimulation [468], in A375-S2 cell lines from melanoma [469], and in SW480 and SW620 cells from colonic tumors [470]. Since silibinin is a potent autophagic stimulator within a range of cell types, its potential to have an anti-HCC impact through this action merits further investigation.

One of the most significant oncological issues requiring resolution is the rise in cancer resistance to chemotherapeutic agents, radiotherapy, or targeted interventions. Polyphenolic substances are able to stimulate apoptosis and autophagy, thus promoting malignant cell death. As dietary components, their intrinsically diminished toxicity, facilitation of lower dosages, and adverse events compared with man-made agents denotes them as low risk. Admixing them with treatments sanctioned by the Food and Drug Administration (FDA) may offer de novo approaches for malignancy therapy and a counter strategy against the significant issue of drug resistance.

Numerous papers have examined the natural phytopolyphenol, curcumin, in terms of its antitumorigenesis and therapeutic properties [471]. Administration of curcumin to Huh7 cells precipitated early autophagy, as evidenced by autophagic vacuolar evolution [472]. In male Sprague–Dawley rodents, survival statistics for thioacetamide-induced HCC were improved following curcumin by autophagy signalling pathway stimulation via protein expression and apoptosis suppression [473]. Adriamycin (doxorubicin) and curcumin together induce autophagy associated with a raised prevalence of autophagosomes in treated juxtanuclear cells. Research evaluating the autophagy suppressor, 3-MA, has also demonstrated the value of combination therapy [474].

Rottlerin, or mallotoxin, is derived from the monkey-faced tree, Mallotus phillippinensis [475]. Numerous signalling pathways and cellular mechanisms contribute to autophagy and the consequent cellular demise stimulated by this compound. Nevertheless, cellular setting, heightened threshold or resistance to apoptosis and stimulated or suppressed signalling pathways are probably the key elements dictating cellular destinies. Rottlerin and associated analogues may be utilized in the configuration of new therapies for autophagy induction in prostate [476] and pancreatic [477] tumors.

Present in soy items, genistein is a naturally arising isoflavonoid noted to exhibit antitumor activities, including the ability to promote cellular demise via apoptosis [478] and autophagy [479]. The latter arises owing to alterations in apoptotic signalling and is advantageous against malignancy cell resistance to chemical agents [480]. During stress, conditions of poor nutrients or a lack of growth factor, genistein can safeguard the cytokeratin matrix. Several researchers have demonstrated its value in surmounting the deleterious consequences of the potent autophagic inhibitor, okadaic acid, on cytoskeletal and cytokeratin configurations in rodent liver cells, which is of note as the latter contributes to autophagic evolution. Despite its autophagic and apoptotic properties in tumor cells, genistein has been shown to have inherently poor oral bioavailability owing to metabolic enzymes and efflux transporters. This issue merits additional study in order to enhance its effectiveness for the therapy of apoptotic-resistant neoplasia [481].

Quercetin is a natural flavonoid molecule found in fruits, vegetables, leaves, and grains. It has antitumor effects related to its ability to target key molecules, organelles, and tumorigenic pathways [482]. Quercetin mediates extensive autophagy and subsequent death in cancer cells by inhibiting proteasome activity [480]. Many studies have shown that quercetin has an effect on autophagy [483,484].

Resveratrol, a natural polyphenol, did not show autophagic response at low concentrations (10 pg/mL); however, at a higher concentrations (20 pg/mL), it activated autophagic cell death in Huh 7 cells [485]. Resveratrol may be effective therapy in apoptosis-resistant ovarian cancer as its acute exposure induces cell death through autophagy in five ovarian cancer cell lines. [403,482,486]. Resveratrol exhibited an effect against human hepatocellular carcinoma (HCC) by inducing autophagy [487]. Resveratrol enhanced the expression of several tubulin subunits that is important for autophagosomes movement inside the cell. Furthermore, there is evidence that resveratrol triggers autophagic death in the cells of chronic myeloid leukemia [488,489]. Trincheri et al. [490] reported that autophagy can be induced with acute exposure to resveratrol.

Autophagy induced in human glioma cells by resveratrol has the ability to inhibit resveratrol-induced apoptosis [491]. Autophagy inhibitors may have the potential to enhance resveratrol antitumor efficacy [491] because autophagy delayed apoptosis and protected the cells from death.

Persistent human papillomavirus infection may stabilize an anti-autophagy factor called ATAD3A, inhibit cell apoptosis in addition to autophagy, and increase drug resistance in uterine cervical cancer. Resveratrol’s antitumor activity was confirmed by its ability to reduce ATAD3A expression and to increase the numbers of autophagosomes [492].

Resveratrol activates autophagic cell death in human prostate cancer PC3 and DU145 cells by downregulating matrix-interacting molecules 1 (STIM1) expression leads to the induction of endoplasmic reticulum stress, which activates AMPK and inhibits the AKT/mTOR pathway [493]. Resveratrol increases the autophagy and autophagy-mediated degradation of p62 in non-small lung adenocarcinoma A549 cells [494].

Various techniques have shown that resveratrol can induce autophagy in breast cancer stem cell-like cells [495]. Resveratrol inhibits tumor cell proliferation and induces apoptosis and autophagy in T acute lymphoblastic leukemia [496]. Silencing of SIRT1 expression inhibits autophagy by inhibiting the phosphorylation of p70RS6K and 4E-BP1, while molecular events are reversed in the presence of resveratrol [497].

A recent review provides more details about resveratrol activity in cancer [498]. Further clinical studies are important to fully evaluate the activity of resveratrol in the killing of tumor cells via autophagy.

Several studies have shown that non-flavonoid and flavonoids polyphenols including quercetin, apigenin, and epigallocatechin gallate (EGCG) can induce autophagy, both in vitro and in vivo.

Members of the phytochemical flavonoid class of compounds, anthocyanins include cyanidin, delphinidin, pelargonidin, and petunidin. Naturally arising pigments, the various subgroups are defined by the flavylium B-ring. Delphinidin leads to notable LCE II lipidation, a cue necessary for autophagosome development [499].

A naturally occurring substance, hydroxycinnamates or E-[6’-(5’-hydroxypentyl) tricosyl]-4-hydroxy-3-methoxycinnamate (EHHM), is acquired from Livistona chinensis. It has been reported that autophagy enhances cell longevity in HCC cells administered EHHM; EHHM could therefore be a potentially efficacious strategy for treatment of this tumor [500].

When berberine is used in combination with lung cancer radiotherapy, in addition to inducing autophagy cell death in vitro and in vivo, it has also been shown to induce autophagy and apoptotic cell death in different hepatocellular carcinoma cells [501]. As evidenced by increased autophagosome formation, LC3B modification, and mitochondrial destruction, berberine enhances non-small cell lung cancer A549 cells’ radiosensitivity undergo autophagy [502]. Berberine derivatives have been shown to induce autophagy, in addition to inhibiting the proliferation of different human colon cancer cells [486,503,504,505,506].

One of the major bioactive components in green tea is Epigallocatechin-3-gallate (EGCG) [507,508,509]. EGCG enhanced the effect of cisplatin and oxaliplatin-induced autophagy in human colorectal cancer cells [510]. EGCG increases the formation of autophagosomes, increasing lysosomal acidification, and stimulating autophagic flux in hepatic cells in vitro and in vivo [511]. Treatment of human hepatocellular carcinoma Hep3B cells with doxorubicin significantly increased a number of autophagic vesicles and levels of autophagic protein markers in tumor cells [512]. This effect resulted in 45% decrease in doxorubicin-induced cell death supporting pro-survival role for autophagy in these experimental conditions [512]. However, EGCG was found to inhibit autophagic signaling, and promoted cellular growth inhibition [512]. The doxorubicin-induced autophagy was blocked by the combination therapy with EGCG [512]. Rapamycin, an autophagic agonist, markedly inhibited the anticancer effect of doxorubicin or its combination with EGCG treatment [512]. Interestingly, EGCG was observed to increase non-apoptotic cell death in human hepatocellular carcinoma cells, cervical cancer cells, and mesothelioma cells [513,514].

Curcumin is the major bioactive component extracted from *Curcuma longa* L., *Curcuma zedoaria* (Christm) Rose., *Curcuma amada* Roxb., and *Curcuma petiolate* [353,358,477]. Curcumin has been found to inhibit cell proliferation in several cancer types through inducing autophagy [515,516,517,518,519,520,521]. For example, in malignant glioma cells curcumin induced G_2_/M arrest and autophagy [480].

Antioxidant N-acetyl-L-cysteine (NAC) blocked the curcumin-induced molecular effects, suggesting that curcumin-induced ROS implicated in autophagosome development [521]. NAC also abolished curcumin-induced activation of ERK1/2 and p38MAPK [521].

The curcumin-induced autophagy was shown to be ROS-dependent [522]. Curcumin induced differentiation of glioma-initiating cells that are responsible for the initiation and recurrence of glioblastoma [523]. Curcumin was found to induce autophagy in these cells in vitro and in vivo [523]. Moreover, curcumin also suppressed tumor formation on intracranial implantation of glioma-initiating cells into mice [523]. Tetrahydrocurcumin, a major metabolite of curcumin, exhibited the antiproliferative effects on human promyelocytic leukemia HL-60 cells by increasing acidic vascular organelle formation specific for autophagy [524].

The survival rate against thioacetamide-induced HCC was observed to be increased by curcumin [473]. The combination of doxorubicin with curcumin caused autophagy stimulation. In the cells treated with this synergistic combination, high levels of the autophagosomes were detected [474].

Curcumin is the most extensively studied natural compound for the prevention and treatment of cancer [471]. However, curcumin has limited therapeutic effect due to its poor bioavailability and effectiveness. Therefore, many curcumin derivatives have been manufactured to evaluate its antitumor potency and those analogs might have an improved autophagy activity compared to curcumin and could provide a better activity toward the HCC cells [471].

Some of the Chinese medicinal herbs induce autophagy [353,366,473]. Fangchinoline triggered autophagy in some of human hepatocellular carcinoma cells. This herb is isolated from Fangji, Stephenia tetrandra S. Moore [525]. Blocking fangchinoline-induced autophagy process markedly modulated the apoptotic pathway [525]. Ginsenosides, major pharmacological active ingredients in Ginseng, induced cell death of tumor cells, thereby improving sensitivity of tumor cells to chemotherapy [329,526,527,528,529,530]. Ginsenosides Rg3 and Rh2 can inhibit cancer cell growth, while Rg3 is instrumental in combating tumor cell resistance to cancer chemotherapy [329,531]. A combination of Rg3 with docetaxel, paclitaxel, cisplatin, or doxorubicin enhanced the sensitivity of prostate cancer and human colon cells to chemotherapy [531]. The anticancer function of ginsenosides is associated with its ability to regulate autophagy in various human cancer cells [329,532,533,534]. Ginsenoside K activated an autophagy pathway mediate by increased autophagic flux [534].

Terpenoids are the largest class of natural compounds exhibiting multiple antitumor properties, especially due to their selectivity toward tumor and cancer stem cells (CSC) [358,535,536]. Sesquiterpene lactones are 15- C terpenoids, such as parthenolide, artemisinin, and thapsigargin, were shown to be beneficial in cancer clinical trials [537,538,539,540]. Parthenolide was found to induce autophagy in the triple-negative breast cancer MDA-MB231 cells [541]. Parthenolide treatment of human hepatocellular carcinoma HepG2 cells resulted in autophagic cell death [542].

Triptolide is a diterpenoid from the roots of Tripterygium wilfordii and was found to inhibit the proliferation of 60 US National Cancer Institute cancer cell lines [543,544]. Furthermore, its anticancer activities were confirmed in various animal models grafted with human tumors resulting in the development of several more water-soluble and less toxic derivatives that entered clinical trials [544]. Triptolide was shown to prevent human pancreatic tumor cell growth both in vitro and in vivo [545,546]. Betulinic acid is a triterpenoid isolated from the bark of the white birch tree that exhibited ant-tumor characteristcs against several cancer cells in vitro and in vivo [547,548]. Betulinic acid and its derivatives were found to decrease the phosphorylation of AKT and induce autophagic cell death in human glioblastoma cells [549]. However, in human multiple myeloma KM3 cells, betulinic acid treatment inhibited autophagy and induced apoptosis [550]. Betulinic acid was recently found to induce death in human cervical cancer HeLa cells, while caspase inhibitors and necrostatin-1 (NEC-1) blocked apoptosis and necroptosis, but not cell death in HeLa cells, implicating caspase-independent mechanisms of cell death in these cells [551].

Oridonin, a diterpenoid extracted from Rabdosia rubescent, promoted autophagy in L929 cells through p38 MAPK and nuclear factor kappa B (NF-kB) pathways [552,553,554]. Terpinen-4-ol was shown to induce autophagic and apoptotic cell death in human promyelocytic leukemic HL-60 cells through inducing the accumulation of LC3B-I/-II, ATG5, and BECN1 proteins cytochrome C released from mitochondria, and decreasing BCL-xL expression [555]. Celastrol, an active compound extracted from the root bark of Tripterygium wilfordii Hook F., exhibited 20S proteasome inhibitor activity, while inducing apoptosis and autophagy in cancer cells [556,557,558,559,560,561]. Both apoptotic and autophagic pathways were found intertwined upon celastrol treatment, since inhibition of apoptosis enhanced autophagy, while suppression of autophagy diminished apoptosis in human osteosarcoma cells [557]. Celastrol could induce hypoxia-inducible factor (HIF)-1a protein accumulation leading to transcriptional activation of HIF-1 target genes [558,559].

Celastrol induced autophagy in human gastric cancer AGS and YCC-2 cells [561]. Moreover, gastric tumor burdens were reduced by celastrol administration in a mouse model with a grafted human gastric tumor [561].

Sulforaphane is found in cruciferous vegetables, such as broccoli, cabbage, cauliflower, and hoary weed [535]. Sulforaphane exhibited antiproliferative properties toward various human tumor cells via various molecular mechanisms [536,562,563,564,565,566]. Sulforaphane was shown to induce the formation of autophagosome-like structures as well as acidic vesicular organelles in human prostate cancer PC-3 cells [536]. Upon sulforaphane exposure, tumor cells showed LC3B-II puncta associated with autophagosomes [536]. Sulforaphane was also shown to disrupt the BCL-2/BECN1 interaction leading to the autophagic pathway initiated by liberated BECN1 [536]. Sulforaphane decreased the phosphorylated AKT-Ser473 level, and simultaneous treatment of sulforaphane with autophagy inhibitors, 3-MA, or chloroquine enhanced drug cytotoxicity and inhibited tumor cell proliferation [536,566].

Recently, a novel alkaloid called Monanchocidin A (MonA) was isolated from the marine sponge Monanchora pulchra [567]. MonA exhibited cytotoxic activity towards human genitourinary cancer cells, including hormone-sensitive and castration-resistant prostate carcinoma cell lines, cisplatin-sensitive and -resistant germ cell tumor cell lines, and different bladder carcinoma cell lines. Whereas, nonmalignant cells were notably less susceptible [567]. MonA was found to induce autophagy and lysosomal membrane permeabilization in cancer cells [567]. Cryptotanshinone and dihydrotanshinone, two lipophilic tanshinones from a traditional Chinese medicine Salvia miltiorrhiza, were shown to induce autophagic flux and LC3B-II accumulation in multidrug-resistant colon cancer cells SW620 and Ad300 cells [568]. Cardamonin is derived from Alpinia katsumadai Hayata (Zingiberaceae) [568]. Cardamonin inhibited cell proliferation and enhanced autophagy in human colon colorectal carcinoma HCT-116 cells [569].

Cannabinoids promote autophagy-dependent apoptosis in melanoma cells [570]. Treatment with A (9)- Tetrahydrocannabinol (THC) activated autophagy, loss of cell viability, and apoptosis, whereas co-treatment with chloroquine prevented THC-induced cell death and autophagy in vitro [570].

Seriniquinone, isolated from a marine bacterium of the genus Serinicoccus, demonstrated potent anti-proliferative activity toward melanoma cell lines by activation of autophagocytosis, while targeting small protein, dermcidin [571]. Oblongifolin C (OC) is a natural small compound extracted from Garcinia yunnanensis Hu. OC is a potent inhibitor of autophagic flux [572,573,574].

Plant lectins have been considered as possible antitumor drugs because of their ability to induce autophagic cell death [575,576,577,578,579,580]. Polygonatum odoratum lectin (POL), from traditional Chinese medicine herb, is a mannose-binding GNA-related lectin and was shown to exhibit apoptosis-inducing and anti-proliferative activities toward a variety of tumor cells [575]. POL could induce both autophagy and apoptosis in NSCLC A549 cells and human breast cancer MCF-7 cells [575,576].

Thymoquinone exposure resulted in caspase-independent, autophagic cell death in human LoVo colon cancer cells [581]. Honokiol (HNK), purified from the Magnolia officinalis bark, is a promising anticancer agent against prostate cancer cells in vitro and (PC-3 xenografts) in vivo [582]. Mollugin, a bioactive phytochemical isolated from Rubia cordifolia L., exhibited anticancer activity against various cancer cells [583].

Jujuboside B is a saponin from the Zizyphus jujuba var. spinosa seeds. Jujuboside B induced autophagy and apoptosis in human AGS and HCT-116 gastric adenocarcinoma cells in vitro and efficiently inhibited cancer growth in a nude mouse xenograft model bearing HCT-116 cells in vivo [584]. Moreover, jujuboside B induced autophagy indicated by the formation of cytoplasmic vacuoles and LC3B-I/II conversion [584]. Bafilomycin A1, which is an autophagy inhibitor, reduced the viability of jujuboside B-induced cells [584]. Feroniellin A (FERO), a novel furanocoumarin, was shown to induce autophagy as well as showing association with LC3B-I to LC3B-II conversion, induction of GFP-LC3B puncta, enhanced expression of BECN1, and ATG5, and inactivation of mTOR in etoposide-resistant human lung carcinoma A549RT-eto cells [585].

A polymethoxy flavonoid, nobiletin has been shown to inhibit cellular replication in human SKOV3/TAX cells through apoptosis and autophagic flux suppression. Comparatively, via Akt signalling, the disturbed autophagic process activated nobiletin-induced apoptosis in this cell line. These data provided evidence to suggest that in human ovarian tumor cells, nobiletin can surmount multi-drug resistance by inhibiting autophagic disruption via Akt modulation [586]. EGFR and TKIs were the most significant for late-stage NSCLC.

Nevertheless, T790M mutation, which increases TKIs resistance generated by EGFR, has transpired to be a key difficulty in tumor therapy. An admixture of wogonin and icotinib was noted to surmount icotinib resistance arising via T709M [587]. An elevated population of intracellular autophagosomes, conversion of LC3B-I to LC3B-II, and Beclin-1 and phosphorylated mTOR expression amplification were identified following the combined administration of wogonin and icotinib. This implies that the two compounds have mutually potentiating influences on cell replication and could play a role in apoptosis and autophagy in EGFR T790M-mutated lung malignancy [587].

Derived from Sophoraflavescens Aiton, matrine is a major quinolizindine alkaloid [588]. The serial signal transduction giving rise to apoptosis from autophagy by triggering p53 has been the subject of discourse [589]. Metabolomic analysis of HepG2 cells administered matrine has revealed lipid droplet metabolites which form macroautophagy substrates partially responsible for immune response activation and apoptosis [590]. Matrine has also been shown to diminish glutathione (GSH) titres; a raised GSH concentration is implicated in tumor resistance to chemotherapy [591].

Mixing autophagy interventions with molecular targeted treatment is thought to offer a positive therapeutic approach to HCC [592]. Data amassed from the last twenty years have emphasized the significance of autophagy in a spectrum of human pathologies. Influencing this process by targeting certain modulators in the core autophagy pathways could therefore impact various pathophysiologies. Malignancies exhibit heightened and diminished autophagic cues, consistent with their cancer-suppressing and promoting characteristics during tumorigenesis. In order to design de novo drugs, recognition of key targets of the autophagic process is essential.

Polyphenols have a distinct capacity to inhibit cell replication and to initiate apoptosis or autophagy in HCC; this ability has drawn attention to their possible use for targeted treatments. In brief, polyphenols, e.g., apigenin, oroxylin A, and resveratrol, may act as inhibitors for the PI3K/Akt/mTOR signalling mechanism. LC3 II evolution, and thus autophagy inducement, is contributed to by luteolin, isoorientin, quercetin, kaempferols, curcumin, adriamycin (doxorubicin) with curcumin, EGCG, EHHM, delphinidin, EF25-(GSH)_2_, oroxylin A, resveratrol, and kaempferols. mTOR signalling pathway phosphorylation may be suppressed by myricetin. Autophagy could be instigated by galangin via the TGF-p receptor/Smad. Safeguarding autophagy linked with the negative modulation of CD147 and ER stress could be provoked by baicalein. Wogonin, in combination with sorafenib, WZ35, and tangeretin has anti-HCC effects mediated through autophagy inhibition.

Multiple biological properties of the naturally occurring polyphenols in the diet encompass antitumor and autophagy-enhancing influences. Research has demonstrated that polyphenolics enact their anti-HCC activities through interference with the autophagic process, e.g., activation of Beclin-1, Atg5, Atg7, Atg9, Atg12, LC3-II, and SQSTM1, together with the modulation of PI3K/Akt/mTOR, PTEN, P38/PPAR-a, JNK/Bcl-2, ER stress, p62, p53-dependent, TGF-p receptor/Smad signaling, and YAP. These data suggest that they are possible agents with multiple modes of action against HCC, which is a catastrophic pathology.

Key factors to take into account with pharmaceutical agents sanctioned by the FDA, e.g., sorafenib, are their adverse event and strength profiles, as these indicate potentially serious unwanted side effects, such as liver toxicity, inflammation, bleeding, fistulas, rashes, high blood pressure, ischemia, and wound reparation issues. Similarly, outcome data on HCC therapies in individuals with advanced cirrhosis are not accessible. Thus, a preferable option would be to couple autophagy-modulating hepatoprotective polyphenolic compounds, with favorable safety statistics, with anti-HCC agents sanctioned by the FDA in order to offer new treatment approaches encompassing autophagy modulation. Furthermore, the development of de novo therapies for HCC should include such polyphenols in their research.

The development of resistance to pharmaceutical agents decreases their therapeutic impact. As autophagy contributes to cancer advancement, influencing this process using naturally arising substances is a potentially encouraging approach to combat multidrug resistance in malignant cells. However, the lack of testing in in vivo models, therapeutic protocols, and evaluations of phytochemical toxicity weakens the published data from single studies and makes it challenging to predict the effectiveness of substances derived from plants in the therapy of specific forms of neoplasia.

### 5.3. Targeting Oncosis

Oncosis is described by cell lysis and rapid cell and organelle swelling, in addition to membrane permeability. Oncosis is associated with intercellular events involving swelling of the mitochondria, depletion of adenosine triphosphate (ATP), failure of calcium ion homeostasis, activation of certain proteases (such as cathepsins and calpains), disruption of lysosome, and finally rupture of the plasma membrane [265,593].

In addition to chemotherapy, radiation, genetic, or immune therapeutic strategies as well as combinatorial approaches, natural antitumor products with promising safety and efficacy are setting an important stage for the new anticancer treatments [288,329,330,331,332,333,334,450,451,594,595] (Table 2).

Artemisinin, which was extracted and separated from *Artemisia annua* L. (sweet wormwood), has been used as one of the well-known traditional Chinese medicines for many years in the treatment of fevers and chills [596]. In pancreatic cancer and renal cell carcinoma, it has been shown that artemisinin induced oncosis-like cell death [597]. The former cell death occurred with the morphotype characteristics of oncosis, while the latter was via the generation of reactive oxygen species (ROS) and the depletion of ATP [597,598,599]. In gastric cancer, artemisinin stimulated cell oncosis by reducing the expression of Vascular endothelial growth factor (VEGF) and increasing the amount of calcium and the expression of calpain-2 [599,600].

The exact mechanism of action of artemisinin is still controversial, and the target of action has not been completely revealed. Current research shows that artemisinin uses multi-approaches and multi-links to influence the tumor progression. Both apoptotic and non-apoptotic cell death are involved in the anticancer activity of artemisinin. Furthermore, artemisinin affects cancer metabolism and immunosuppression. However, the related literature is still limited, and more in-depth research into these aspects is required.

### 5.4. Targeting Methuosis

One of the most recent forms of non-apoptotic cell death is methuosis. The name of methuosis, which is derived from the Greek ‘methuo’ and means ‘to drink to intoxication‘, was chosen because the most prominent characteristic in cells undergoing this phenotype of death is the accumulation of large fluid-filled cytoplasmic vacuoles that originate from macropinosomes [601,602,603].

Macropinocytosis is defined as a non-selective liquid-phase endocytic pathway for the extracellular substances’ uptake [604]. Macropinocytosis was first recognized in 1931, and the used term was pinocytosis, or cell-drinking [605]. Then, in 1986–1992, the stimulation of membrane ruffling and fluid-phase pinocytosis were described [606,607] and a review about macropinocytosis was published in 1995 [608]. Later, it was found that abnormal expression of the RAS genes in glioma cells and gastric cancer would cause cellular degeneration in addition to vacuolization [609], which belongs to micropinocytosis. Eventually, this led to a new distinct phenotype of cell death. It was discovered that abnormal genetic manipulations as well as trace amounts of certain drugs can stimulate methuosis in cancer cells [610,611,612].

Recently, the relationship between tumors and macropinocytosis has attracted more attention as it might has a great research value for tumor survival and treatment [604]. Accordingly, several researchers are interested in macropinocytosis as a new target for cancer treatment in addition to its potential for antitumor drugs delivery and the design of antitumor drugs that can induce methuosis or abrogate the process of macropinocytosis have been reported. However, some challenges and queries exist about the research in this field.

Firstly, it is not clearly confirmed whether methuosis represents a recent unique phenotype of controlling cell death or whether it is just a subtype of oncosis or necrosis [601,613]. Additionally, the molecular mechanisms related to methuosis remain unclear despite the known importance of RAS genes and more detailed and specific evidence linking macropinocytosis directly to cell death is lacking. Indeed, there are more stimulants of methuosis waiting to be discovered for clinical use.

Additionally, macropinocytosis can both enhance cancer survival and has detrimental effects on cancers [614,615,616,617,618,619,620,621,622,623,624,625]. This makes methuosis different from classical apoptosis and other non-apoptosis death forms. Perhaps this means that macropinocytosis has a “threshold” between enhancing cancer cells survival and death. However, more studies are needed to know what exactly this “threshold”.

The understanding of molecular pathways involved in non-apoptotic cell death induced by natural anticancer drugs would assist in exploiting novel molecular targets of plant-derived compounds necessary to advance safer and effective anticancer therapeutics allowing to circumvent cancer drug resistance [293,334,341,449,480,503,507,524,610,626,627,628,629,630,631,632]. The success with the natural product shikonin, which was able to induce multiple cell death mechanisms, supports the notion that natural compounds could bypass specific drug resistance developed by tumor cells using simultaneous activation of multiple death pathways [293,334,341,449,480,503,507,524,610,626,627,628,629,630,631,632]. Therefore, a reasonable combination of several cell death inducers that complement each other will maximize their efficacy while minimizing their side effects [293,334,341,449,480,503,507,524,610,626,627,628,629,630,631,632]. Although in vitro studies have shown that natural compounds have a strong ability to induce non-apoptotic death of tumor cells, more in vivo studies are needed to strengthen this concept before entering clinical applications [610,629,630,631,632,633]. Figure 3 summarizes the main natural products and their mechanism of action in triggering non-apoptotic cell death.

**Table 2 biomedicines-09-01353-t002:** A list of natural compounds with an effect on non-apoptosis cell death in tumor cells.

Compound Name	Target	Reference
Matrine	Necroptosis	[344]
Neoalbaconol	Necroptosis	[345]
Shikonin	Necroptosis Autophagy	[293,322,334,341,346,449,480,503,507,524,610,626,627,628,629,630,631,632]
Emodin	Necroptosis	[348]
Ungeremine	Necroptosis	[349]
Staurosporine	Necroptosis	[362,363,364]
Obatoclax	Necroptosis	[365,366,367]
Piperlongumine	Necroptosis	[369,370]
Eupomatenoid-5	Necroptosis	[371]
Rottlerin	Autophagy	[456,634]
Genistein	Autophagy	[635,636]
Quercetin	Autophagy	[483,611]
Resveratrol	Autophagy	[458,480,492,494,612,637,638,639,640,641,642,643,644,645,646,647] [458,480,494,612,637,638,639,640,641]
Anthocyanins	Autophagy	[499]
Hydroxycinnamates	Autophagy	[500]
Berberine	Autophagy	[486,501,503,504,505,594,595,648,649]
Epigallocatechin-3-gallate	Autophagy	[534,538,539,650]
Curcumin	Autophagy	[515,517,518,519,520,522,523,651,652,653,654]
Fangchinoline	Autophagy	[525]
Ginsenosides	Autophagy	[329,526,527,528,529,530,531,532,533]
Terpenoids	Autophagy	[358,535,536]
Triptolide	Autophagy	[568,570,655]
Betulinic acid	Autophagy	[547,548]
Oridonin	Autophagy	[558,559]
Celastrol	Autophagy	[561,656]
Sulforaphane	Autophagy	[581,585,587,592]
Monanchocidin A	Autophagy	[567]
Cryptotanshinone	Autophagy	[568,569]
dihydrotanshinone	Autophagy	[568,569]
Cannabinoids	Autophagy	[570]
Seriniquinone,	Autophagy	[571]
Oblongifolin C	Autophagy	[572,573,574]
Polygonatum odoratum lectin	Autophagy	[575,576]
Honokiol	Autophagy	[582,583]
Jujuboside B	Autophagy	[607,609,613]
Nobiletin	Autophagy	[587]
Matrine	Autophagy	[589,590,591]
Parthenolide	Autophagy	[565,657,658]
Allicin	Autophagy	[659,660]
Citreoviridin	Autophagy	[659,660,661,662]
7-hydroxydehydronuciferine	Autophagy	[663]
Glycyrrhetinic acid	Autophagy	[664]
Honokiol	Autophagy	[579]
Artemisinin	Oncosis	[597,600]
Matrine	Necroptosis	[368]
Neoalbaconol	Necroptosis	[369]
Shikonin	Necroptosis Autophagy	[317,346,358,365,370,473,527,532,551,652,653,654,656,659,660,661,662,663]
Emodin	Necroptosis	[372]
Ungeremine	Necroptosis	[373]
Staurosporine	Necroptosis	[386,387,388]
Obatoclax	Necroptosis	[389,390,391]
Piperlongumine	Necroptosis	[393,394]
Eupomatenoid-5	Necroptosis	[395]
Rottlerin	Autophagy	[480,665]
Genistein	Autophagy	[666,667]
Quercetin	Autophagy	[507,668]
Resveratrol	Autophagy	[482,516,518,653,669,670,671,672,673,674,675,676,677,678,679,680] [482,518,653,669,670,671,672,673,674]
Anthocyanins	Autophagy	[523]
Hydroxycinnamates	Autophagy	[524]
Berberine	Autophagy	[525,527,528,529,530,623,624,681,682]
Epigallocatechin-3-gallate	Autophagy	[534,538,539,650]
Curcumin	Autophagy	[542,544,545,546,547,549,550,683,684,685,686]
Fangchinoline	Autophagy	[552]
Ginsenosides	Autophagy	[353,553,554,555,556,557,558,559,560]
Terpenoids	Autophagy	[358,535,536]
Triptolide	Autophagy	[568,570,655]
Betulinic acid	Autophagy	[571,572]
Oridonin	Autophagy	[582,583]
Celastrol	Autophagy	[585,687]
Sulforaphane	Autophagy	[581,585,587,592]
Monanchocidin A	Autophagy	[593]
Cryptotanshinone	Autophagy	[265,594]
dihydrotanshinone	Autophagy	[265,594]
Cannabinoids	Autophagy	[595]
Seriniquinone,	Autophagy	[596]
Oblongifolin C	Autophagy	[597,598,599]
Polygonatum odoratum lectin	Autophagy	[600,601]
Honokiol	Autophagy	[607,608]
Jujuboside B	Autophagy	[607,609,613]
Nobiletin	Autophagy	[615]
Matrine	Autophagy	[617,618,619]
Parthenolide	Autophagy	[565,657,658]
Allicin	Autophagy	[688,689]
Citreoviridin	Autophagy	[688,689,690,691]
7-hydroxydehydronuciferine	Autophagy	[692]
Glycyrrhetinic acid	Autophagy	[693]
Honokiol	Autophagy	[604]
Artemisinin	Oncosis	[626,628]

## 6. Clinical Studies

Although most antitumors reduce the size of tumors significantly [665], they fail to remove them completely. As a result, the tumor resists treatment and relapses. The main reason for cancer treatment failure by chemotherapy is the emergence of tumor cell death resistance to drugs during cancer progression, thus representing a central issue in chemotherapeutic approaches to this tumor. Multidrug resistance (MDR) is one of the major clinical challenges in malignancy treatment and compromises the effectiveness of conventional antitumor drugs.

The ability of natural products and their derivatives to prevent, inhibit, and reverse the progression of cancer has been clinically studied. Surveys indicate that approximately 80% of cancer patients use natural products in combination with classic anticancer drugs [666]. This shows that many cancer patients are very interested in using natural products as nutritional supplements or complementary or alternative medicines. They hope that these natural products can significantly reduce the side effects caused by anticancer drugs, enhance the immune response, and enhance the effectiveness of anticancer drugs. Some people believe that they will actually prevent or reverse the progression of cancer.

More and more evidence has shown that natural compounds are highly specific to tumor cells and have minimal adverse effects on normal neighboring cells; therefore, they bring great hopes for future anticancer therapies [334,633]. It is reported that more than 3000 plant species can treat cancer. Thus far, about 30 plant-derived compounds have been isolated and tested in cancer clinical trials [334] and are used currently in clinical practice exhibiting advantageous results, with some exhibiting serious toxic side effects.

The principal objectives for combination therapies encompassing prolongation of survival rates and enhancing life quality include mitigating the cytotoxic adverse event profiles of pharmaceutical agents and simultaneously diminishing tumor resistance and unwanted drug effects.

Natural compounds such as flavonoids, sesquiterpenes, alkaloids, diterpenoids, and saponins, in addition to polyphenolic compounds, to overcome drug resistance [667,668] are substituted or applied in combination with existing drugs. The principal objectives for combination therapies encompassing prolongation of survival rates and enhancing life quality include mitigating the cytotoxic adverse event profiles of pharmaceutical agents and simultaneously diminishing tumor resistance and unwanted drug effects. These natural compounds are known to have anticancer effects and can both kill cancer cells and restore drug sensitivity.

Although sufficient data were available from clinical trials conducted on animals to prove the efficacy of resveratrol with respect to tumor therapy, few in vitro clinical studies have been conducted in human cells. Thus, there is a dearth of results relating to human trials that assess the effectiveness of resveratrol in cancer resistance treatment. The data exhibited unpredictable outcomes with respect to the use of resveratrol because the majority of these clinical trials used different doses and routes of administration with a small sample size of patients [498].

Resveratrol can efficiently exhibit its antitumor effects in combination with other chemotherapeutic agents in addition of having an excellent safety profile. A number of important challenges need to be considered before bringing resveratrol to the bedside owing to its rapid metabolism giving rise to limited bioavailability in humans [498].

The role of melatonin in cancer treatment and prevention has been widely studied, and numerous experimental studies proved the antitumor effect of melatonin against many types of cancers. The combination of conventional anticancer therapies with melatonin showed promising results through reinforcing the therapeutic effects of these therapies [669]. Overall, the high safety profile, diverse mechanisms of action, and high efficiency of melatonin support its use in cancer prevention and treatment [669].

It has been proposed that melatonin’s benefit in mitigating the toxic effects of chemotherapy and its association with aberrant mitochondrial function should be explored using double-blind, placebo-controlled trials. It can be expected that a plethora of information will emerge over the next ten years relating to the way in which melatonin exerts a positive effect in conjunction with chemotherapeutic agents [670]. The development of resistance to therapy, together with the occurrence of tumor spread, means that the investigation of de novo modes of treatment for malignancies is essential.

It is well established that therapy for individuals presenting with glioblastomas is complex; curative surgery is nearly impossible, and the majority of tumors exhibit a high recurrence rate despite treatment with radiation and chemotherapy [671]. Thus, several workers have concentrated on the development of de novo adjuvant treatment approaches, favoring natural products in order to offer anticancer agents that are suitable for the clinical practice. A number of studies have documented the characteristics of melatonin with respect to glioblastomas. Melatonin has well-known anti-oxidant actions, and its antitumor effects are becoming acknowledged. It therefore has potential to thwart the resistance to numerous anticancer agents that plagues treatment of glioblastomas [671,672,673]. Melatonin was reported as a candidate to overcome the multi-drug resistance glioblastoma treatment [672,694,695]. Additional work is required to design novel molecular products, combination treatments, and optimal dosing regimens. Although a few studies have reported anticancer actions of melatonin in relation to glioblastoma in vitro, as yet, few animal models have been published, and there is scant literature available on this subject in humans.

Lissoni et al. published a study investigating treatment with an admixture of melatonin and aloe vera [674]. The purpose of this trial was to determine whether these two products could act in synergy to enhance the anticancer characteristics of melatonin. Fifty patients with malignancy, including patients who had developed resistance to chemotherapy, radiation, and hormone treatments, or who were unable to tolerate chemical anticancer agents, were recruited for the study. Eight weeks after therapy commencement, no effect on lesion regression was seen in the cohort only receiving melatonin. In the group taking aloe vera and melatonin, 2/24 patients (8%) exhibited a partial response. The safety profile for melatonin was benign [674].

The data relating to the utilization of melatonin as an adjunct to chemotherapy are encouraging, both in terms of augmenting the effectiveness of therapy and mitigating adverse event profiles [672,675]. However, clinical studies that have investigated the clinical efficacy of melatonin in conjunction with other forms of treatment in patients with neoplasia, excluding glioblastoma, have usually been performed outside evidence-based recommendations following lack of success with conventional therapy and a guarded life expectancy.

Traditional Chinese Medicine (TCM) uses a combinatorial method of two or more agents to achieve a synergistic effect [676]. TCM has been utilized worldwide as a complementary or an alternative medicine and has long been used to treat cancer in China [677,678,679,680,681,682]. Examples of TCM’s main components include alkaloids, flavonoids, and saponins. Numerous natural products originating from TCM can reverse multidrug resistance [650]. Research has shown that flavonoids reverse multi-drug resistance [683].

Curcumin is a common term used for a mixture of curcuminoids that are purified from the Indian spice turmeric powder, mainly comprised of curcumin (curcumin I), demethoxycurcumin (curcumin II), and bisdemothoxycurcumin (curcumin III) [684]. Curcumin is a traditional medicine and the main curcuminoid of Curcuma longa [685,686].

Curcuminoids are known to have many biological activities, including anti-inflammatory [655,687], anticancer [657], and antiviral properties [658,688]. Moreover, both curcumin and its major metabolite tertrahydrocurcumin were found to restore drug sensitivity in tumor cells overexpressing the MOR-linked ABC transporters Pgp [684,689], MRP1 [689,690], and ABCG2 [689,691] by directly inhibiting their functions. More recently, curcumin was found to be active against MDR tumors in mice as well [692]. Considering its inhibitory effect on multiple ABC drug transporters and its many beneficiary biological properties, it is not surprising that curcumin has become one of the most exciting natural product modulators in recent years.

Clinically, poor bioavailability is the one major problem with using curcumin. The levels of curcumin in plasma and tissues remain low after oral consumption, reported to be in the range of nanomolars and picomolars, respectively [693]. Curcumin is lipophilic and very insoluble in nature, and is also rapidly metabolized in the intestine and excreted in the urine, which means that high doses of curcumin must be consumed for it to be biologically relevant and effective [696]. For instance, in one study, the level of curcumin was only barely detectable in human plasma even after a dose of curcumin as high as 12.0 g [697]. Therefore, several approaches have been investigated to improve the delivery of curcumin in the human system [698], including the use of liposomal curcumin [699], curcumin nanoparticles [700,701], curcumin phospholipid complex or structural analogues of curcumin [702], or the use of a combination of curcumin and piperine. Piperine has been shown to block the metabolism of curcumin by P450 A3 and by other hepatic and intestinal pathways involved in glucuronidation of this compound [703]. In addition, it was observed that piperine increased the bioavailability of orally given curcumin in both rats and humans with no adverse effect [703]. Thus, it should be useful to test whether the combination of curcumin and piperine improves the bioavailability of co-administered antitumor drugs in cancer patients.

Clinical trials offer the opportunity to verify and to identify the impact, side effects, and pharmacokinetics of therapeutic agents. Since curcumin’s bioavailability is low, many curcumin formulations have been manufactured and have undergone testing in clinical trials [694,695,704]. A phase I clinical study was performed in order to establish the safety and pharmacokinetic profiles of theracurmin in individuals with malignancy of the pancreas and biliary tract in whom conventional chemotherapeutic agents had been unsuccessful [695]. Daily theracurmin, in combination with chemotherapy founded on gemcitabine, was administered. No additional side effects or rise in their incidence were reported. A phase II pilot study showed promising data for the admixture of docetaxel/prednisone and curcumin in individuals with prostate tumors resistant to orchidectomy. In 59% and 40%, respectively, either complete or some degree of prostate-specific antigen response was noted. This work offered further data indicating a high rate of response and acceptability for curcumin administration during treatment for malignancy [705]. Despite the optimistic published articles and clinical trials regarding curcumin’s potential effectiveness against cancer, there is evidence to show that curcumin has no therapeutic benefits [706]. However, researchers still think that because of suggestive trends in trial results and because curcumin can interact with many proteins, there is still justification for further study [707]. The antitumor activity of curcumin remains unconfirmed until better experiments are carried out.

More importantly, both phase I and II clinical studies with curcumin have been carried out and showed some encouraging results. Despite its poor bioavailability, phase I studies indicated that curcumin is well tolerated [708] and provided substantial improvement in patients with advanced colorectal cancer when treated with curcumin (360–500 mg) [693], with minimal drug–curcumin interactions [709]. Likewise, phase II studies showed treatment with curcumin can improve the clinical outcome in patients with advanced pancreatic cancer [710]. These clinical studies suggest that it would be worthwhile to test curcumin as an adjuvant along with traditional chemotherapy drugs to overcome MDR in cancer patients.

Among all natural product modulators, the most well-studied and well-known are flavonoids, which include flavonols, flavones, isoflavones, flavanols, flavanolols, flavanones, and chalcones [711]. Typically, each person consumes a substantial amount of flavonoids per day from fruits, vegetables, food supplements, and tea. They are known to have many prominent health benefits [712,713], including anticancer properties [714,715,716]. In terms of MDR, flavonoids have been studied and characterized extensively by many research groups to determine their capability to restore drug sensitivity in MDR tumor cells [717,718,719,720].

Artemisinin possesses some advantages, including less susceptibility to resistance, that makes it worthy of development as a novel anticancer agent. In Pubmed, there are only three studies in the last three years. Some previous clinical studies before 2019 have been comprehensively reviewed by Efferth and Zhang et al. [721,722]. Despite using different modes of artemisinin administration for different cancers, no solidly concluded results can be seen yet.

Sanctioned by the State FDA of China, the compound Kushen Injection (CKI) has been utilized as an adjunctive therapy to Western antitumor medication for various forms of malignancy [723]. It comprises alkaloids, flavonoids, saccharides, and organic acids [724] and is derived from the medicinal plants, Radix Sophorae flavescentis and Rhizoma Smilacis Glabrae [725]. CKI is believed to be efficacious in preventing metastasis and overcoming multi-drug resistance [725]. However, to date there are no studies to this effect in the literature available in English; any existing research presents few convincing results. Thus, in vivo work and the clinical pertinence of CKI requires additional study in order to determine the efficacy of any chemosensitizing properties.

Polyphenolic natural products, such as Ellagic Acid and Schisandrins, represent a chemically unique class of molecules as potential anticancer agents to overcome multidrug resistance in cancer [93]. However, clinical studies to support this benefit are required.

Epigallocatechin gallate (EGCG) is one of the major bioactive components in green tea. EGCG enhances the effect of cisplatin and oxaliplatin-induced resistance of cancer cells and exerts synergistic effects with anticancer agents. In addition to cisplatin and oxaliplatin, these agents include temozolomide, doxorubicin, resveratrol, vardenafil, erlotinib, and curcumin [726,727,728].

In order to appraise the acceptability, pharmacokinetics, and effectiveness of EGCG in humans for tumor therapy, clinical studies are in progress. A phase I clinical trial evaluating therapy for radiation dermatitis in patients with breast neoplasia studied concurrent radiotherapy and EGCG solution. The highest dose of 660 pM of EGCG was without significant side effects [729], and the solution was deemed efficacious for the therapy of radiation dermatitis. EGCG was also appraised in a phase II clinical study to explore its advantages in acute radiation-induced oesophagitis in individuals with stage 3 lung tumors. Oral EGCG delivery was proven to be useful; phase III studies are expected to follow [730].

Gambogic acid (GA) is one of the main compounds derived from the gambogic resin exuded from a plant of the genus Garcinia [731]. It has a variety of biological activities, including anticancer properties [732]. The combination of GA and other anticancer agents has been widely used to improve the therapeutic effectiveness against various tumors [732,733,734,735]. Cisplatin resistance, which is a main clinical challenge in the treatment of lung cancer, can be decreased in human NSCLC cisplatin-resistant A549/ DDP cells by combining cisplatin and GA [734].

To asses the safety and effectiveness of GA in patients with advanced malignant tumors, different doses have been compared in a phase IIa clinical trial [736]. GA is safe at a dose of 45 mg/m^2^. Patients taking GA on days 1–5 within a two-week cycle showed a higher rate of disease control, with only grade I and II adverse reactions. A phase IIb clinical trial with a larger sample size of participants would be required to better investigate the safety and efficacy of GA.

A major limitation in all of the clinical studies has been the identification of appropriate pharmacodynamic biomarkers evaluating changes in autophagy.

Autophagosomal configurations can act as scaffolds to initiate apoptosis [365] and necroptosis [365,737], and so their accrual may enhance these cues within some contexts. If this notion were true, it may be preferable to inhibit autophagosomal breakdown with a lysosomal inhibitor instead of suppressing autophagosome production which may avoid malignant cell eradication. Lastly, the issue of autophagy inducers to circumvent oncogenesis requires addressing. It has been postulated that enhancing autophagy may hinder the onset of neoplasia by restricting genomic mutations, fostering oncogene-produced quiescence and diminishing cancer-induced inflammation [738]. This is therefore a complicated problem owing to the interplay of autophagy with various genetic contexts, e.g., the mutations of p53 recognized in tumors of the pancreas [739] and breast [740] where the reaction to autophagy initiation may be impacted by the sequencing of p53, thus creating a pro- or anti-carcinogenic effect.

Contemporary clinical trial designs frequently permit specimen gathering from malignancies, together with serum pre and post therapy. These may assist in the generation of improved bioindicators to act as pharmacodynamic markers for the effectiveness of autophagy inhibitors and to enhance patient selection for this therapy type. If enhanced clinical trials are added to in-depth molecular and cellular appraisal in order to comprehend the pathways underpinning the setting-reliant influence of autophagy on malignancy, a more logical foundation to inform judgements about when and in which trajectory autophagy could be influenced during tumor treatment could be developed. As changes in autophagy are inevitable during malignancy and such alterations will impact cancer progression, turning a blind eye to the issue is a poor choice. It is preferable to comprehend the biology and then use that information in effectively designed clinical studies.

The final objective of laboratory tests is to attain clinical usage. In clinical trials, subjects are typically split into two cohorts, i.e., controls administered traditional chemotherapy and the intervention arm who are additionally given natural products. The impact of Fritillariae thunbergia was evaluated in 90 individuals with acute leukemia [741]; compared with controls, individuals in the intervention arm exhibited a smaller population of leukemic cells within the bone marrow, diminished MDR1 protein titres, and a lower remission rate. The same substance was appraised in 30 individuals with acute leukemia with elevated MDR1 expression [742]. The intervention arm subjects had three-fold reduced MDR1 protein concentrations, an increased response rate to therapy, i.e., 55% as opposed to 20%, and a smaller bone marrow proportion of leukemic cells, i.e., 26% compared to 50%. A total of 36 individuals with acute leukemia underwent measurement of bone marrow cell mRNA titres of MRP and ^2M using real-time PCR. An MRP/^2M parameter ≥0.3 was deemed MRP positive [743]. Those subjects receiving a 15-day course of 120 mg daily intravenous ligustrazine had a greater likelihood (45.5%) of becoming negative for MRP than controls (7.1%) and to have a reduced fraction of bone marrow leukemic cells, i.e., 21.4% as opposed to 55.6%.

Inhibitors or modulators derived from natural resources are occasionally termed ‘fourth-generation inhibitors’. Such substances offer a spectrum of de novo chemical scaffolds that are apposite for the creation of new agents. It can be anticipated that scientists appreciate the role of screening for novel natural compounds that exhibit these properties, as they are more likely to have a positive outcome than many existing products. There is an enormous range of resources that could be used; biologically active constituents are currently derived from vegetation, fungi, and even sea life, following which they are purified and studied in depth. An advantage is that these natural substances are typically of minimal toxicity and produce few side effects.

Individuals with solid tumors [744] and disseminated breast malignancy [745,746] found artemisinin to be tolerable and non-toxic. Nevertheless, prompts for ongoing surveillance during artemisinin prescription to record adverse effects should be contemplated particularly when it is utilized in high dosages. If appropriate, pharmaceutical agents used to avoid side effects should be given together. Furthermore, despite the fact that in patients with malignancy it can be challenging to distinguish whether adverse incidents are a result of pathology or medication, recent data indicate that such reports may have been associated with artemisinin; this should be evaluated in further clinical studies. Certain variables, i.e., route of delivery, quantity, and course length of medication, affect safety and effectiveness, and this needs to be studied in depth. Lastly, the accessible data from the phase I clinical studies for artemisinin were restricted, and the patient population was low in number. Thus, more expansive clinical trials for phases II, III, and IV are required in order to offer more compelling data for the appropriateness of artemisinin in clinical cancer therapy.

Although uncertainty remains, clinical modulation of autophagy in oncology is currently in progress [747] with most endeavors aimed towards autophagy suppression. In fact, a survey of the ClinicalTrials.gov website, entering the terms ‘autophagy’ and ‘cancer’, brought up over 50 studies which concentrated on autophagy inhibition and assessment to enhance clinical endpoints. In keeping with other sectors of tumor biology, e.g., the promise of the immune system to potentiate and to suppress carcinogenesis and advancement, the principal factor impacting positive outcomes from treatments relating to autophagy arises from the delineation of the way in which the autophagic process influences cancer onset and evolution.

Chemical biology methods and cell culture-based approaches are powerful but have limitations that can unintentionally introduce confusion or uncertainty conclusion. Consideration of these limitations may help avoid common pitfalls and, in the fullness of time, lead to useful reinterpretation of existing data. Thus, additional future studies using in vivo systems and better clinical trials for the clinical management of resistance to drug-induced tumor cell death are needed to determine the clinical effectiveness and safety of these natural compounds.

## 7. Safety Aspects of Natural Products in Cancer Management

While many natural products used for cancer are associated with minimal or no risk, this is not true for all such therapies. Potential toxicities include direct toxicity of natural products, indirect effects of natural products due to interactions with other medications, and also the risk to the patient who uses natural products to avoid or delay established, effective treatment in the management of cancer disease [748,749].

There are some potential side effects associated with commonly used natural products and other types of CAM [750,751]. For example, green tea can cause emesis and diarrhea, in addition to insomnia and confusion.

Many natural products are pharmacologically active, raising concerns about potential interactions with conventional therapy, both cytotoxic agents, and other medical therapies [752,753,754,755]. Many anticancer drugs are metabolized through the cytochrome p450 system. Thepolyphenols present in green tea suppress many cytochrome p450 enzymes, which are important in drug metabolism and induce other drug-metabolizing enzymes. Several components of green tea and green tea extract can antagonize the effectiveness of bortezomib by different interaction mechanisms while increasing the effect of medicine such as anthracycline, taxanes, and tamoxifen (CYP3A4 inhibition) [756]. Additionally, Essiac, which consists of multiple biologically active substances, can act synergistically with anticancer drugs by the cytotoxic or immunosuppressive activities of anthraquinones existing in this mixture or through its inhibition of CYP3A [757]. Ginkgo biloba and panax ginseng enhance the functional activity of many drug-metabolizing CYP family enzymes, and it is recommended to be avoided in patients receiving agents metabolized by CYP3A4 or CYP2C19 [758]. Curcuma, used in some types of cancers, can cause nausea, gastric irritation, diarrhea, and bleeding problems. Curcuma interacts with many drugs, mainly doxorubicin and cyclophosphamide [759].

Although not a direct “toxic” effect, the use of natural products may result in a significant delay in instituting conventional treatment that is of documented benefit for a specific condition [760]. As non-natural products, in some cases, are imagined to negate the benefit of the therapy, it is usually required that natural products alone be taken for the duration of the CAM treatment. This strategy of care can lead to the rejection of effective medical therapies [748]. Although the data are limited, there is an adverse impact of refusing/delaying standard treatments in favor of alternative therapies [761,762].

In a previous report, 258 patients diagnosed with nonmetastatic breast, prostate, lung, and colorectal cancer in the National Cancer Database between 2004 and 2013 who underwent alternative medicine treatment as the sole therapy (identified as those coded as “other unproven cancer treatments administered by nonmedical personnel”) and who also did not receive conventional cancer treatment were compared with a matched cohort of 1032 patients who received conventional cancer therapy [762]. Patients who chose alternative therapy had higher refusal rates for surgery (70 versus 0.1 percent), radiation therapy (53 versus 22 percent), chemotherapy (34 versus 3 percent), and hormone therapy (34 versus 3 percent). Alternative medicine use was associated with worse five-year overall survival (82.2 versus 86.6 percent), while the use of alternative treatment was independently associated with a greater risk of death (hazard ratio [HR] 2.08, 95% CI 1.50-2.90). These data indicate that the refusal of conventional cancer therapy was associated with the mortality risk. Important limitations of this study are its observational nature and the reliance on medical diagnosis coding at a single facility, which may have underascertained the use of conventional cancer treatment for patients who received treatment at a different facility or those who initially received alternative medicine prior to presenting to the facility that was reporting data.

The use of autophagy-related kinase inhibitors/activators may lead to unwanted and uncontrolled side effects, despite their potential therapeutic benefits in animal models. Considering the protective properties of autophagy on neurons, it is reasonable to enhance the brain specificity of autophagy-related therapies for neurodegenerative conditions. Different delivery approaches and photodynamic chemotherapy are proposed to attain the goal of organ specificity. Additionally, as known for the common kinase drugs, these autophagy-related kinase inhibitors/activators share the complications of target selectivity and resistance [763].

The lack of cancer cells’ killing selectivity of thapsigargin prevent its direct use as an anticancer agent. To transport thapsigargin directly to cancer sites, some prodrugs have been developed. For example, G115 and G202 are prodrugs created by a conjugation of thapsigargin to substrates of proteolytic enzymes that are available only in tumor cells. Additionally, JQ-FT is an antileukemic prodrug established by a conjugation of a thapsigargin derivative and folic acid. These prodrugs represent an efficient way to overcome thapsigargin cytotoxicity and provide targeted cancer therapy [764].

## 8. Conclusions

Natural products are emerging as a promising source for effective anticancer agents. The numerous sources of these products cause high diversity in targets and mechanisms of action. Such diversity has encouraged scientists to consider natural products as therapies for drug resistance in cancer. Some natural products showed high potential to target drug resistance mechanisms in cancer and caused tumor regression. Many of these natural compounds were successfully considered as therapies in preclinical and clinical studies. However, the use of natural product as a standard therapy to treat drug resistance is still limited. Further studies are needed to explore the potential of natural products in combination therapies to overcome drug resistance. Future studies can focus on studying the possible synergistic effect of natural product combinations to target multiple biomarkers in drug resistance. Studies can also consider using natural products as adjuvant treatments to augment conventional anticancer therapies.

## Figures and Tables

**Figure 1 biomedicines-09-01353-f001:**
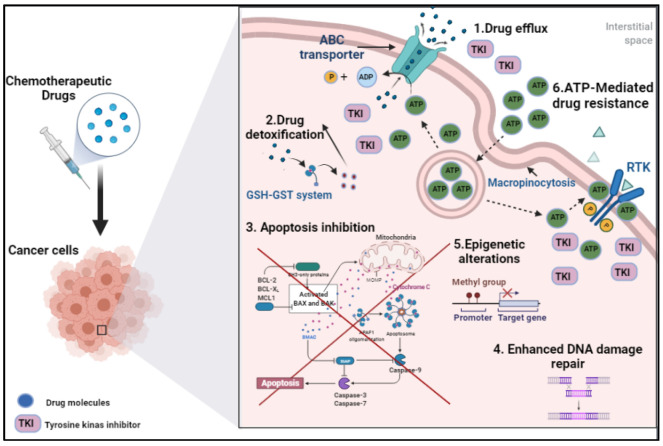
Illustration of drug chemoresistance mechanisms in cancer cells [1]. ABC: ATP binding cassette, RTK: receptor tyrosine kinase, TKI: tyrosine kinase inhibitors.

**Figure 2 biomedicines-09-01353-f002:**
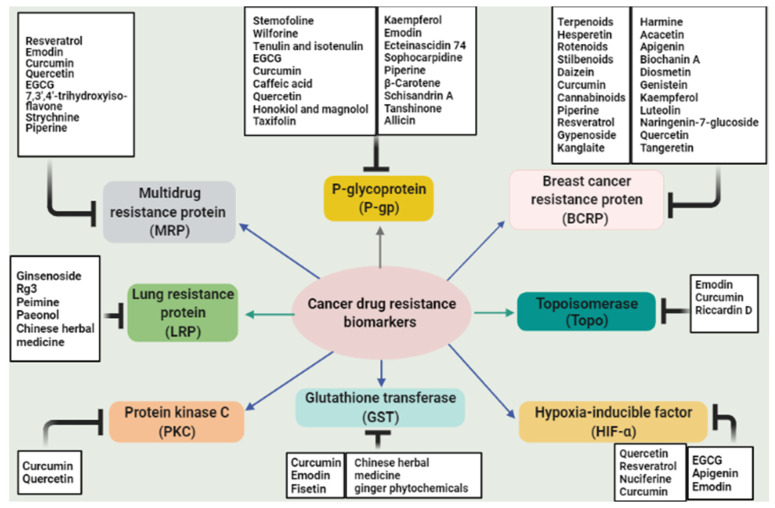
Summary of the main natural compounds targeting multidrug resistance biomarkers in cancer.

**Figure 3 biomedicines-09-01353-f003:**
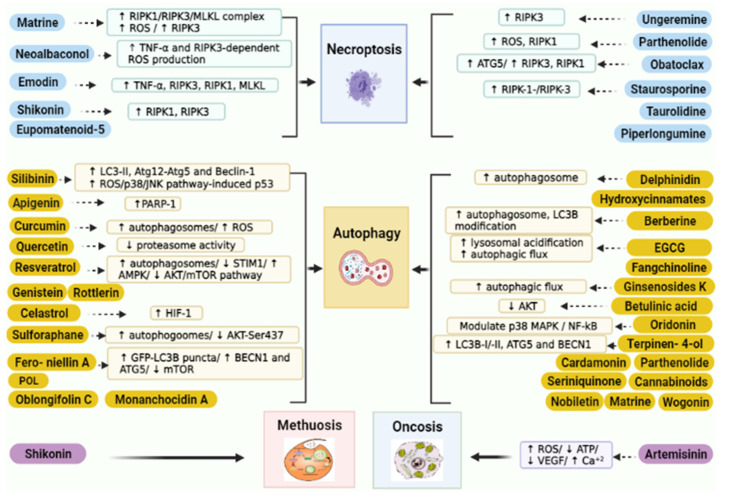
Natural products mechanism of action in promoting non-apoptotic cell death in cancer cells. RIPK1, receptor-interacting serine/threonine-protein kinase 1; MLKL, mixed lineage kinase domain-like; ROS, reactive species; TNF-α, tumor necrosis factor; ATG5, autophagy related 5; LC-II, light chain 2; STIM1, stromal interaction molecule 1; AKT, protein kinase B; mTOR, mammalian target of rapamycin; AMPK, AMP-activated protein kinase; HIF-1, hypoxia-inducible factor-1; BECN1, beclin1; NF-kB, nuclear factor kappa B.

## Data Availability

Not applicable.

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
