# Peer review of "Targeting Drug Chemo-Resistance in Cancer Using Natural Products"

_biomedicines, 2021, doi:10.3390/biomedicines9101353_

Round 1

Reviewer 1 Report

This manuscript is  a very comprehensive review, covering wide range of nautral products used for anti-cancer therapies. It is clearly written and very well strucutred. 

Molecular basis of drug chemo-resistance in cancer  are very well explained. Targets of natural products in cancer chemo-resistance are listed exhaustively and examples of natural products targeting those elements is given. I would like to see however  more emphasis givnet on the Phorbol Esters and other Diterpenoid classes modulating PKC activity reviewd f.ex extesively by Simon Remy and Marc Litaudon (molecules 2019) -  this is for 3.5. Protein kinase C paragraph.

Next section, "Targeting non-apoptotic cell death using natural products is also well strucutred" and plethora of natural product targeting necrosis and apoptosis of cancer cells is given.  I think more citation is needed for artemisinin mode of action (lines 1416-1421). 

Final section about Clinical studies reqiures somehow more emphais on the negative side of develpent of natural products as drugs. Plenty of example of severe cytotoxicity of secondary plant metabolites are know and those often prevent entry of the compounds showing promissing in vitro activties into clinical trials. There are well documented side-effects of PKC modulators from animal studies (disruption of brain-blood barrier) that can be given as examples. Mechanisms of targeting the natural products direclty into cancer cells or chemcial derivatisation could be given as a way of ovecoming cytotoxic effects. Thapsigargin can be given as the example here.

Finally, fully synthetic anti-cancer chemotherapeutc agents, derived from chemical libraries should be mentioned somewhere as an alternative to natural products. 

Author Response

Thank you for your positive feedback.

The manuscript was subjected to extensive revision and all changes were tracked. All your comments were considered in the revised manuscript.

We hope that the modified manuscript will meet your expectations

Reviewer 2 Report

This is very interesting review paper dealing with several possibilities to target cancer with natural products. I have some minor comments-

  1. Central message in the abstract is missing specially concluding remarks.
  2. It would be nice to include a pathways (sketch diagram) in section 3 and section 4 detailing the mechanism of drug targeting in cancer. Including pathways in the text, not only improve the quality of  manuscript but also attract readership.
  3. Author should add few lines as "future perspectives" in conclusion section.

Author Response

Thank you for your positive feedback.

The manuscript was subjected to extensive revision and all changes were tracked. All your comments were considered in the revised manuscript and we believed that your comments add more value to our work.

We hope that the modified manuscript will meet your expectations
